# Bézier Flow: a Surface-wise Gradient Descent Method for Multi-objective Optimization

**Akiyoshi Sannai**                                                    *sannai.akiyoshi.7z@kyoto-u.ac.jp*
*Kyoto University*
*National Institute of Informatics*

**Yasunari Hikima**                                                    *hikima.yasunari@fujitsu.com*
*Fujitsu Limited*
*Kyushu University*

**Ken Kobayashi**                                                    *kobayashi.k@iee.eng.isct.ac.jp*
*Institute of Science Tokyo*

**Akinori Tanaka**                                                    *akinori.tanaka@riken.jp*
*RIKEN AIP*
*RIKEN iTHEMS*
*Keio University*

**Naoki Hamada**                                                    *hamada-n@klab.com*
*KLab Inc.*

**Reviewed on OpenReview:** *https://openreview.net/forum?id=I1gALvbRxj*

## Abstract

This paper proposes a framework to construct a multi-objective optimization algorithm from a single-objective optimization algorithm by using the Bézier simplex model. Additionally, we extend the stability of optimization algorithms in the sense of Probably Approximately Correct (PAC) learning and define the PAC stability. We prove that it leads to an upper bound on the generalization error with high probability. Furthermore, we show that multi-objective optimization algorithms derived from a gradient descent-based single-objective optimization algorithm are PAC stable. We conducted numerical experiments with synthetic and real multi-objective optimization problem instances and demonstrated that our method achieved lower generalization errors than the existing multi-objective optimization algorithms.

## 1 Introduction

A multi-objective optimization problem is a problem to seek a solution which minimizes (or maximizes) multiple objective functions $f_1, \ldots, f_M : X \to \mathbb{R}$ simultaneously over a domain $X \subseteq \mathbb{R}^L$:

$$\text{minimize} \quad \boldsymbol{f}(\boldsymbol{x}) \coloneqq (f_1(\boldsymbol{x}), \ldots, f_M(\boldsymbol{x}))^\top$$
$$\text{subject to} \quad \boldsymbol{x} \in X \subseteq \mathbb{R}^L.$$

Each objective function can have a different optimal solution, so we need to consider the trade-off between two or more solutions. Therefore, the notion of Pareto ordering is taken into consideration, which is defined by

$$\boldsymbol{f}(\boldsymbol{x}) \prec \boldsymbol{f}(\boldsymbol{y}) \stackrel{\text{def}}{\iff} f_m(\boldsymbol{x}) \leq f_m(\boldsymbol{y}) \text{ for all } m = 1, \ldots, M,$$
$$\text{and } f_m(\boldsymbol{x}) < f_m(\boldsymbol{y}) \text{ for some } m = 1, \ldots, M.$$

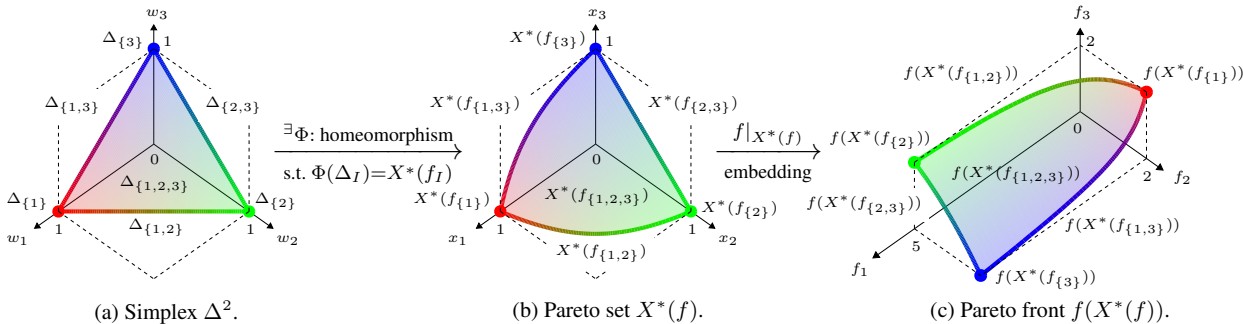

(a) Simplex $\Delta^2$.     (b) Pareto set $X^*(f)$.     (c) Pareto front $f(X^*(f))$.

Figure 1: A simplicial problem $f = (f_1, f_2, f_3)^\top : \mathbb{R}^3 \to \mathbb{R}^3$. An $M$-objective problem $f$ is *simplicial* if the following conditions are satisfied: (i) there exists a homeomorphism $\Phi : \Delta^{M-1} \to X^*(f)$ such that $\Phi(\Delta_I) = X^*(f_I)$ for all $I \subseteq \{1, \ldots, M\}$; (ii) the restriction $f|_{X^*(f)} : X^*(f) \to \mathbb{R}^M$ is a topological embedding (and thus so is $f \circ \Phi : \Delta^{M-1} \to \mathbb{R}^M$).

In multi-objective optimization, the goal is to obtain the Pareto set and Pareto front, which are respectively defined as:

$$X^\star(\boldsymbol{f}) := (\boldsymbol{x} \in X \mid f(\boldsymbol{y}) \nprec f(\boldsymbol{x}) \text{ for all } \boldsymbol{y} \in X), \; \boldsymbol{f}X^\star(\boldsymbol{f}) := (\boldsymbol{f}(\boldsymbol{x}) \in \mathbb{R}^M \mid \boldsymbol{x} \in X^\star(\boldsymbol{f})).$$

The Pareto set/front usually has an infinite number of points, whereas most of the numerical methods for solving the problem give us a finite set of points as an approximation of the Pareto set/front (e.g., goal programming (Eichfelder, 2008; Miettinen, 1999), evolutionary computation (Deb, 2001; Deb & Jain, 2013; Zhang & Li, 2007), homotopy methods (Harada et al., 2007; Hillermeier, 2001), and Bayesian optimization (Hernandez-Lobato et al., 2016; Yang et al., 2019)). Such a finite-point approximation cannot reveal the complete shape of the Pareto set and Pareto front. In addition, the finite-point approximation suffers from the "curse of dimensionality" since the dimensionality of the Pareto set and Pareto front is $M - 1$ in generic problems (see Wan (1977; 1978) for a rigorous statement). With this background, we focus on an optimization algorithm to obtain a parametric hypersurface describing the Pareto set.

There is a common structure of the Pareto set/front across a wide variety of problems, which can be utilized to enhance approximation. In many problems, obtained solutions imply the Pareto set/front is a curved $(M - 1)$-simplex, e.g., airplane design (Mastroddi & Gemma, 2013), hydrologic modeling (Vrugt et al., 2003), PI controller tuning (Reynoso-Meza et al., 2015), building design (Safarzadegan Gilan et al., 2016), motor design (Contreras et al., 2016), and Lasso's hyper-parameter tuning (Hamada & Ichiki, 2020). To mathematically identify such a class of problems, Kobayashi et al. (2019) defined the *simplicial* problem (see Figure 1). Hamada et al. (2020) showed that strongly convex problems are simplicial under mild conditions, which implies that facility location (Kuhn, 1967) and phenotypic divergence modeling in evolutionary biology (Shoval et al., 2012) are simplicial. Kobayashi et al. (2019) showed that the Pareto set and Pareto front of any simplicial problem can be approximated with arbitrary accuracy by a Bézier simplex.

By using this advantage of the Bézier simplex model, we propose a novel strategy to construct a multi-objective optimization algorithm from a single-objective optimization algorithm. With a given single objective optimization algorithm, such as a gradient descent method, this scheme updates the Bézier simplex to obtain the Pareto set. In addition, we analyze the theoretical properties of the multi-objective optimization algorithm derived from our scheme. Specifically, we define Probably Approximately Correct (PAC) stability as an extension of the stability of optimization algorithms and prove that the PAC stability leads to an upper bound on the generalization gap in the sense of PAC learning. Our contributions are summarized as follows:

1. We devise a strategy to construct a multi-objective optimization algorithm from a single-objective optimization algorithm with the Bézier simplex. Unlike most of the existing multi-objective optimization methods, the algorithm derived from our scheme has the advantage of obtaining a parametric hypersurface that represents the Pareto set of a given simplicial, Lipschitz continuous, differentiable multi-objective optimization problem to be solved.

2. We define PAC stability, which is an extension of the stability introduced by Hardt et al. (2016) to the PAC learning settings, and show that PAC stability gives an upper bound on the generalization gap with a high probability. Also, we prove that when we employ a gradient-based optimization algorithm as a single optimization algorithm, the derived multi-objective optimization algorithm is PAC stable.

3. We conducted numerical experiments and demonstrated that the multi-objective optimization algorithm constructed by our scheme achieved lower generalization errors than the existing multi-objective optimization algorithm. In addition, the algorithm given by our scheme can efficiently obtain the Pareto set with a small number of sample points.

**Related Work**    Kobayashi et al. (2019) proposed Bèzier simplex fitting algorithms, the all-at-once fitting, and inductive skeleton fitting to describe Pareto fronts, and Tanaka et al. (2020) analyzed the asymptotic risk of the fitting algorithms. The two fitting algorithms focus on post-optimization processes and assume that we have an approximate solution set of the Pareto set in advance. Thus, these algorithms alone cannot solve multi-objective optimization problems. Recently, Maree et al. (2020) proposed a bi-objective optimization algorithm that updates the Bézier curve. However, this algorithm exploits the structure of the bi-objective optimization problem and can not be applied when the number of objective functions is more than or equal to three. To the best of our knowledge, we are the first to propose a general framework of multi-objective optimization with the Bézier simplex and show its theoretical properties.

## 2  Preliminaries

### 2.1  Probability simplex

Let $[M] = \{1, \ldots, M\}$ be a set of $M$ points. We consider the set of probability distribution $\boldsymbol{t} \in \mathbb{R}^M$ over $[M]$. The set of probability distributions over $[M]$ is equal to the simplex

$$\Delta^{M-1} := \left\{ (t_1, \ldots, t_M)^\top \in \mathbb{R}^M \; \middle| \; t_m \geq 0, \; \sum_{m=1}^M t_m = 1 \right\}.$$

Let $C(X)$ be the space of continuous functions over $X$, and we define the function $F \colon [M] \to C(X)$ by $F(m) = f_m$. Then, we have the expectation function

$$\mathbb{E}(\boldsymbol{f}) \colon \begin{array}{ccc} \Delta & \longrightarrow & C(X) \\ \cup & & \cup \\ \boldsymbol{t} & \longmapsto & \mathbb{E}_{\boldsymbol{t}}(F) \end{array}.$$

Furthermore, if $f_m$ is strongly convex for all $m \in [M]$, then the following function is well-defined:

$$\operatorname{argmin} \mathbb{E}(\boldsymbol{f}) \colon \begin{array}{ccc} \Delta & \longrightarrow & X \\ \cup & & \cup \\ \boldsymbol{t} & \longmapsto & \operatorname{argmin} \mathbb{E}_{\boldsymbol{t}}(F) \end{array}.$$

Note that $\mathbb{E}_{\boldsymbol{t}}(F) = \sum_m t_m f_m$ follows from the definition. $\mathbb{E}_{\boldsymbol{t}}(F)$ corresponds to the sum of a function chosen continuously along $\boldsymbol{t}$ from $\boldsymbol{f}$. As a direct consequence from Theorem 2 in Mizota et al. (2021), the mapping $\operatorname{argmin} \mathbb{E}(\boldsymbol{f})$ gives a continuous surjection onto $X^\star(\boldsymbol{f})$ if $f_m$ is strongly convex for all $m \in [M]$.

### 2.2  Simplicial problem

A multi-objective optimization problem is characterized by its objective map $\boldsymbol{f} = (f_1, \ldots, f_M)^\top \colon X \to \mathbb{R}^M$. We define the *J-subsimplex* for an index set $J \subseteq [M]$ by $\Delta_J^{M-1} := \{(t_1, \ldots, t_M)^\top \in \Delta^{M-1} \mid t_m = 0 \; (m \notin J)\}$. The problem class we wish to consider is a problem in which the Pareto set/front has the simplex structure. Such a problem class is defined as follows.

**Definition 2.1** (Kobayashi et al. (2019)). A problem $\boldsymbol{f}\colon X \to \mathbb{R}^M$ is *simplicial* if there exists a map $\boldsymbol{\phi}\colon \Delta^{M-1} \to X$ such that for each non-empty subset $J \subseteq [M]$, its restriction $\boldsymbol{\phi}|_{\Delta_J^{(M-1)}}\colon \Delta_J^{M-1} \to X$ gives homeomorphisms

$$\boldsymbol{\phi}|_{\Delta^{M-1}{}_J}\colon \Delta_J^{M-1} \to X^\star(\boldsymbol{f}_J), \quad \boldsymbol{f} \circ \boldsymbol{\phi}|_{\Delta_J^{M-1}}\colon \Delta_J^{M-1} \to \boldsymbol{f}X^\star(\boldsymbol{f}_J).$$

We call such $\boldsymbol{\phi}$ and $\boldsymbol{f} \circ \boldsymbol{\phi}$ a *triangulation* of the Pareto set $X^\star(\boldsymbol{f})$ and the Pareto front $\boldsymbol{f}X^\star(\boldsymbol{f})$, respectively.

### 2.3 Bézier simplex

Let $\mathbb{N}$ be the set of nonnegative integers (i.e., $\mathbb{N} \coloneqq \{0, 1, 2, \dots, \}$) and

$$\mathbb{N}_D^M \coloneqq \left\{ (d_1, \dots, d_M)^\top \in \mathbb{N}^M \ \middle|\ \sum_{m=1}^M d_m = D \right\}.$$

For $\boldsymbol{t} \coloneqq (t_1, \dots, t_M)^\top \in \Delta^{M-1}$ and $\boldsymbol{d} \coloneqq (d_1, \dots, d_M)^\top \in \mathbb{N}_D^M$, we denote by $\boldsymbol{t}^{\boldsymbol{d}}$ a monomial $t_1^{d_1} \dots t_M^{d_M}$. The Bézier simplex of degree $D$ in $\mathbb{R}^L$ with control points $\{\boldsymbol{p_d}\}_{\boldsymbol{d} \in \mathbb{N}_D^M} \subseteq \mathbb{R}^L$ is a map $\boldsymbol{b}\colon \Delta^{M-1} \to \mathbb{R}^L$, which is defined by

$$\boldsymbol{b}(\boldsymbol{t} \mid \boldsymbol{P}) \coloneqq \sum_{\boldsymbol{d} \in \mathbb{N}_D^M} \binom{D}{\boldsymbol{d}} \boldsymbol{t}^{\boldsymbol{d}} \boldsymbol{p_d}, \tag{1}$$

where $\binom{D}{\boldsymbol{d}} \coloneqq \frac{D!}{d_1! d_2! \dots d_M!}$ is a multinomial coefficient and $\boldsymbol{P} \in \mathbb{R}^{|\mathbb{N}_D^M| \times L}$ represents a matrix of control points, which is defined as

$$\boldsymbol{P} \coloneqq \begin{pmatrix} (\boldsymbol{p}_1)_1 & (\boldsymbol{p}_1)_2 & \cdots & (\boldsymbol{p}_1)_L \\ (\boldsymbol{p}_2)_1 & (\boldsymbol{p}_2)_2 & \cdots & (\boldsymbol{p}_2)_L \\ \vdots & \vdots & \ddots & \vdots \\ (\boldsymbol{p}_{|\mathbb{N}_D^M|})_1 & (\boldsymbol{p}_{|\mathbb{N}_D^M|})_2 & \cdots & (\boldsymbol{p}_{|\mathbb{N}_D^M|})_L \end{pmatrix}. \tag{2}$$

Define $\boldsymbol{z}(\boldsymbol{t})$ as a vector of a coefficient of the Bézier simplex (1) with respect to control points $\{\boldsymbol{p_d}\}_{\boldsymbol{d}}$, i.e.,

$$\boldsymbol{z}(\boldsymbol{t}) \coloneqq \left( \binom{D}{\boldsymbol{d}_1} \boldsymbol{t}^{\boldsymbol{d}_1}, \dots, \binom{D}{\boldsymbol{d}_{|\mathbb{N}_D^M|}} \boldsymbol{t}^{\boldsymbol{d}_{|\mathbb{N}_D^M|}} \right)^\top \in \mathbb{R}^{|\mathbb{N}_D^M|}.$$

Then, the definition of Bézier simplex (1) is represented as $\boldsymbol{b}(\boldsymbol{t} \mid \boldsymbol{P}) = \boldsymbol{P}^\top \boldsymbol{z}(\boldsymbol{t})$. It is known that a Bézier simplex is a universal approximator of continuous functions (Kobayashi et al., 2019), and thus, the mapping $\operatorname{argmin} \mathbb{E}(\boldsymbol{f})$ can be approximated by Bézier simplices in arbitrary precision. From this theoretical advantage, we construct a general framework to obtain a multi-objective optimization method from a single-objective optimization method with Bézier simplices.

## 3 Proposed Algorithm

A number of methods have been studied in the context of multi-objective optimization. Many of the methods are designed to apply to any multi-objective optimization problem; however, the individual methods are written in separate contexts and are not unified. Therefore, in this paper, we introduce a general framework to obtain a multi-objective optimization method $\mathcal{M}(\mathcal{A})$ from a single-objective optimization algorithm $\mathcal{A}$. Moreover, in contrast to the existing methods, which aim to find a finite set that approximates the Pareto set/front, our proposed method obtains a parametric hypersurface representing the Pareto set/front of multi-objective problems to be solved.

In our proposed algorithm, we aim to obtain control points of a Bézier simplex that represents the Pareto set of a multi-objective problem from a single-objective optimization algorithm $\mathcal{A}$. In this paper, a single-objective

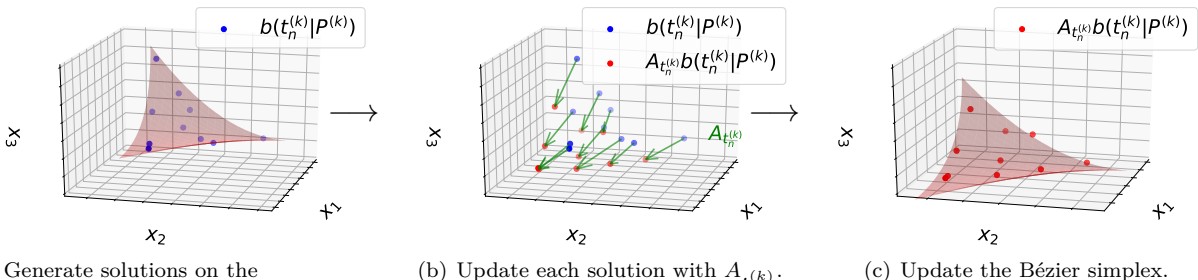

(a) Generate solutions on the Bézier simplex.

(b) Update each solution with $A_{\boldsymbol{t}_n^{(k)}}$.

(c) Update the Bézier simplex.

Figure 2: Conceptual diagram of $\mathcal{M}(\mathcal{A})$ at the $k$th iteration; the red surfaces in (a) and (c) represent the Bézier simplices.

optimization algorithm $\mathcal{A}$ is a map from the direct product of the sample space $\mathcal{Z}$ and the space of loss functions $\mathcal{L}$ to the space of model parameters $\mathcal{W}$, i.e., $\mathcal{A} : \mathcal{Z} \times \mathcal{L} \to \mathcal{W}$. Then, for any $\boldsymbol{t} \in \Delta^{M-1}$, we denote by $A_{\boldsymbol{t}}$ an update rule in $\mathcal{A}$, which is defined by the loss function $\mathbb{E}_{\boldsymbol{t}}(F)$.

Our algorithm begins by setting the initial control points $\boldsymbol{P}^{(1)}$. At the $k$-th iteration ($k \geq 1$), we randomly draw $N$ samples $\{\boldsymbol{t}_n^{(k)}\}_{n=1}^N$ from the uniform distribution on $\Delta^{M-1}$ and obtain $N$ data points $\{\boldsymbol{b}(\boldsymbol{t}_n^{(k)}|\boldsymbol{P}^{(k)})\}_{n=1}^N$ on the Bézier simplex defined by the current control points $\boldsymbol{P}^{(k)}$. Next, we update each $\boldsymbol{b}(\boldsymbol{t}_n^{(k)}|\boldsymbol{P}^{(k)})$ by $A_{\boldsymbol{t}_n^{(k)}}$, i.e.,

$$\boldsymbol{x}_n^{(k)} = A_{\boldsymbol{t}_n^{(k)}}(\boldsymbol{b}(\boldsymbol{t}_n^{(k)}|\boldsymbol{P}^{(k)})). \tag{3}$$

Then, we update the Bézier simplex with the obtained pairs of data $\{(\boldsymbol{t}_n^{(k)}, \boldsymbol{x}_n^{(k)})\}_{n=1}^N$. Specifically, we solve the following least squares problem to fit a Bézier simplex to $\{(\boldsymbol{t}_n^{(k)}, \boldsymbol{x}_n^{(k)})\}_{n=1}^N$:

$$\underset{\boldsymbol{P} \in \mathbb{R}^{|\mathbb{N}_D^M| \times L}}{\text{minimize}} \frac{1}{N} \sum_{n=1}^N \left\| \boldsymbol{x}_n^{(k)} - \boldsymbol{b}(\boldsymbol{t}_n^{(k)} \mid \boldsymbol{P}) \right\|_2^2, \tag{4}$$

where $\boldsymbol{P}$ is a decision variable to be optimized, and $\|\cdot\|_2$ denotes the Euclidean norm on $\mathbb{R}^L$ Let $\boldsymbol{X}^{(k)} \coloneqq (\boldsymbol{x}_1^{(k)}, \boldsymbol{x}_2^{(k)}, \dots, \boldsymbol{x}_N^{(k)})^\top \in \mathbb{R}^{N \times L}$ and $\boldsymbol{Z}^{(k)} \coloneqq (\boldsymbol{z}(\boldsymbol{t}_1^{(k)}), \boldsymbol{z}(\boldsymbol{t}_2^{(k)}), \dots, \boldsymbol{z}(\boldsymbol{t}_N^{(k)}))^\top \in \mathbb{R}^{N \times |\mathbb{N}_D^M|}$ be matrix of $\boldsymbol{x}_n^{(k)}$ and $\boldsymbol{z}(\boldsymbol{t}_n^{(k)})$, respectively. Then, the problem (4) is equivalent to the following problem:

$$\underset{\boldsymbol{P} \in \mathbb{R}^{|\mathbb{N}_D^M| \times L}}{\text{minimize}} \frac{1}{N} \left\| \boldsymbol{X}^{(k)} - \boldsymbol{Z}^{(k)} \boldsymbol{P} \right\|_F^2, \tag{5}$$

where $\|\cdot\|_F$ denotes the Frobenius norm on $\mathbb{R}^{N \times L}$. Since the optimization problem (5) is an unconstrained convex quadratic optimization, and it can be shown that the symmetric matrix $\boldsymbol{Z}^{(k)\top} \boldsymbol{Z}^{(k)}$ is regular with probability 1 (refer to Lemma B.1 and its proof), the update rule for control points is described as follows:

$$\boldsymbol{P}^{(k+1)} = \left( \boldsymbol{Z}^{(k)\top} \boldsymbol{Z}^{(k)} \right)^{-1} \boldsymbol{Z}^{(k)\top} \boldsymbol{X}^{(k)}. \tag{6}$$

We repeat this procedure until $k$ reaches the maximum number of iterations specified by the user. We summarize the pseudocode for solving multi-objective optimization problems with the above procedure in Algorithm 1 and show its conceptual diagram in Figure 2.

## 4 PAC Stability and Generalization Gap

Assume that there is an unknown distribution $\mathcal{D}$ over some space $\mathcal{Z}$. We take $S = (\boldsymbol{t}_1, \dots, \boldsymbol{t}_N)$ of $N$ examples drawn i.i.d. from $\mathcal{D}$. Then, the generalization error is defined by

$$R[\boldsymbol{P}] \coloneqq \mathbb{E}_{\boldsymbol{t} \sim \mathcal{D}}[\ell(\boldsymbol{P} \mid \boldsymbol{t})],$$

---

**Algorithm 1** Multi-objective Optimization Method $\mathcal{M}(\mathcal{A})$ from Single Optimization Method $\mathcal{A}$

---

1: Set $k \leftarrow 1$ and the initial control point $\boldsymbol{P}^{(k)}$.
2: **while** $k \leq K$ **do**
3:    Draw $\{\boldsymbol{t}_n^{(k)}\}_{n=1}^N$ for which each $\boldsymbol{t}_n^{(k)}$ is drawn i.i.d. from the uniform distribution on $\Delta^{M-1}$.
4:    Obtain $\{\boldsymbol{b}(\boldsymbol{t}_n^{(k)}|\boldsymbol{P}^{(k)})\}_{n=1}^N$ by Equation (1).
5:    Obtain $\{\boldsymbol{x}_n^{(k)}\}_{n=1}^N$ by Equation (3).
6:    Update control points by Equation (6).
7:    $k \leftarrow k+1$.
8: **end while**
9: **return** $\boldsymbol{P}^{(K+1)}$.

---

where $\ell \in \mathcal{L}$ is a given loss function, and $\ell(\boldsymbol{P} \,|\, \boldsymbol{t})$ denotes the loss of the model described by $\boldsymbol{P}$ with an input $\boldsymbol{t}$. Since the generalization error cannot be measured directly, we instead consider the empirical error defined by $R_S[\boldsymbol{P}] := \frac{1}{N} \sum_{n=1}^N \ell(\boldsymbol{P} \,|\, \boldsymbol{t}_n)$. Then, the generalization gap of $\boldsymbol{P}$ is defined as the difference between empirical error and generalization error, i.e.,

$$R_S[\boldsymbol{P}] - R[\boldsymbol{P}]. \tag{7}$$

We consider a potentially randomized algorithm $\mathcal{A}$ (e.g., stochastic gradient descent) and the expected value of Equation (7):

$$\mathbb{E}_{\mathcal{A}}[R_S[\mathcal{A}(S)] - R[\mathcal{A}(S)]], \tag{8}$$

where we represent $\mathcal{A}(S)$ as $\mathcal{A}(S, \ell)$ for notational simplicity.

To treat the approximate behavior of the expected value with respect to the sample, we consider the following. First, take an event $C \subset \mathcal{Z}^N$ that has a high probability of occurring. Then, the conditional generalization error under the condition $C$ is defined by:

$$\hat{R}[\boldsymbol{P}] := \mathbb{E}_{(\boldsymbol{t}_1,\ldots,\boldsymbol{t}_N) \sim \mathcal{D}_C^N}\left[\frac{1}{N} \sum_{i=1}^N \ell(\boldsymbol{P} \,|\, \boldsymbol{t}_i)\right],$$

where $\mathcal{D}_C^N$ is the conditional probability distribution of $C$. Note that if $C = \mathcal{Z}^N$, $\hat{R}[\boldsymbol{P}]$ is equal to $R[\boldsymbol{P}]$.

Next, we consider the approximate expected value of Equation (8) by

$$\hat{\mathbb{E}}_S \mathbb{E}_{\mathcal{A}}\left[R_S[\mathcal{A}(S)] - \hat{R}[\mathcal{A}(S)]\right], \tag{9}$$

where $\hat{\mathbb{E}}_S$ is the conditional expected value of $C$. This invariant allows us to discuss the expected value of the generalization gap with respect to events. The following introduces the definition of *probably approximately correct (PAC) uniform stability*. This is a PAC-like expansion of the uniform stability in Hardt et al. (2016).

**Definition 4.1.** A randomized algorithm $\mathcal{A}$ is *PAC uniformly stable* if for any $\varepsilon \in (0,1)$, there exists $\delta > 0$ and an event $D_\varepsilon \subset \mathcal{Z}^{N+1}$ which occurs with probability at least $1 - \varepsilon$ such that

$$\sup_{\boldsymbol{t}} \mathbb{E}_{\mathcal{A}}[|\ell(\mathcal{A}(S) \,|\, \boldsymbol{t}) - \ell(\mathcal{A}(S') \,|\, \boldsymbol{t})|] < \delta, \tag{10}$$

where $S = (\boldsymbol{t}_1, \ldots, \boldsymbol{t}_N)$ and $S' = (\boldsymbol{t}_1, \ldots, \boldsymbol{t}_i', \ldots, \boldsymbol{t}_N)$ are samples differing in at most one example, drawn from $\mathcal{D}$, satisfying $(\boldsymbol{t}_1, \ldots, \boldsymbol{t}_i, \boldsymbol{t}_i', \boldsymbol{t}_{i+1}, \ldots, \boldsymbol{t}_N) \in D_\varepsilon$. Furthermore, a *PAC uniformly stable* randomized algorithm $\mathcal{A}$ is *decomposable* if for any $\varepsilon \in (0,1)$, there are events $B_\varepsilon \subset \mathcal{Z}$ such that $D_\varepsilon = B_\varepsilon^{N+1}$.

With the PAC stability, we show the following, which ensures that if an algorithm is PAC uniformly stable, the difference between its generalization and empirical error is small with high probability.

---

**Algorithm 2** Surface-wise Gradient Descent Method

---

1: Set $k \leftarrow 1$ and the initial control point $\boldsymbol{P}^{(k)}$.
2: **while** $k \leq K$ **do**
3:     Draw $\{\boldsymbol{t}_n^{(k)}\}_{n=1}^N$ for which each $\boldsymbol{t}_n^{(k)}$ is drawn i.i.d. from the uniform distribution on $\Delta^{M-1}$.
4:     Obtain $\{\boldsymbol{b}(\boldsymbol{t}_n^{(k)} \,|\, \boldsymbol{P}^{(k)})\}_{n=1}^N$ by Equation (1).
5:     Obtain $\{\boldsymbol{x}_n^{(k)}\}_{n=1}^N$ by Equations (3) and (11).
6:     Update control points by Equation (12).
7:     $k \leftarrow k+1$.
8: **end while**
9: **return** $\boldsymbol{P}^{(K+1)}$.

---

**Theorem 4.2** (Proof is shown in Appendix A). *Let $\mathcal{A}$ be a decomposable PAC uniformly stable randomized algorithm. Then, for any $\varepsilon \in (0,1)$ and $\delta > 0$ in Theorem 4.1, there exists an event $C_\varepsilon \subset \mathcal{Z}^N$ which occurs with probability at least $1 - \varepsilon$ such that,*

$$\left| \hat{\mathbb{E}}_S \mathbb{E}_{\mathcal{A}} \left[ R_S[\mathcal{A}(S)] - \hat{R}[\mathcal{A}(S)] \right] \right| < \delta,$$

*where $\hat{\mathbb{E}}_S$ is the conditional expected value of $C_\varepsilon$ and $\hat{R}$ is the conditional generalization error under the condition $C_\varepsilon$.*

## 5   A Surface-wise Gradient Descent Method

Next, we discuss the case that an algorithm $\mathcal{A}$ is a gradient descent method. We refer to the method as *a surface-wise gradient descent method* since the method is designed to generate a sequence of hypersurfaces. Here, we employ the gradient descent-based update rule in Equation (3) as follows:

$$
\begin{aligned}
A_{\boldsymbol{t}_n^{(k)}}\left(\boldsymbol{b}(\boldsymbol{t}_n^{(k)} \,|\, \boldsymbol{P}^{(k)})\right) &:= \boldsymbol{b}(\boldsymbol{t}_n^{(k)} \,|\, \boldsymbol{P}^{(k)}) - \alpha^{(k)} \mathrm{d}_{\boldsymbol{x}} f\left(\boldsymbol{b}(\boldsymbol{t}_n^{(k)} \,|\, \boldsymbol{P}^{(k)}) \,\Big|\, \boldsymbol{t}_n^{(k)}\right) \\
&= \boldsymbol{b}(\boldsymbol{t}_n^{(k)} \,|\, \boldsymbol{P}^{(k)}) - \alpha^{(k)} J_{\boldsymbol{f}}\left(\boldsymbol{b}(\boldsymbol{t}_n^{(k)} \,|\, \boldsymbol{P}^{(k)})\right)^\top \boldsymbol{t}_n^{(k)},
\end{aligned}
\tag{11}
$$

where $A_{\boldsymbol{t}_n^{(k)}}$ is the update rule, $\alpha^{(k)} \in (0,1]$ is a step size, $\mathrm{d}_{\boldsymbol{x}}$ is a first derivative with respect to $\boldsymbol{x}$, $f(\cdot \,|\, \boldsymbol{t})$ is a weighted sum of objective functions $f_1, \ldots, f_M$ by $\boldsymbol{t}$, and $J_{\boldsymbol{f}}(\boldsymbol{x})$ is a matrix of gradient of $f_m$ at $\boldsymbol{x}$ defined by $J_{\boldsymbol{f}}(\boldsymbol{x}) := (\nabla f_1(\boldsymbol{x}), \ldots, \nabla f_M(\boldsymbol{x}))^\top \in \mathbb{R}^{M \times L}$. Define $\boldsymbol{B}^{(k)}$ and $\boldsymbol{G}^{(k)}$ as

$$
\boldsymbol{B}^{(k)} := \boldsymbol{Z}^{(k)} \boldsymbol{P}^{(k)}, \quad \boldsymbol{G}^{(k)} := \left( \left(\boldsymbol{t}_1^{(k)}\right)^\top J_{\boldsymbol{f}}\left(\boldsymbol{P}^{(k)\top} \boldsymbol{z}(\boldsymbol{t}_1^{(k)})\right), \ldots, \left(\boldsymbol{t}_N^{(k)}\right)^\top J_{\boldsymbol{f}}\left(\boldsymbol{P}^{(k)\top} \boldsymbol{z}(\boldsymbol{t}_N^{(k)})\right) \right)^\top.
$$

Note that $\boldsymbol{B}^{(k)}$ and $\boldsymbol{G}^{(k)}$ are variables determined by $\{\boldsymbol{t}_n^{(k)}\}_{n=1}^N$ which is drawn i.i.d. from the uniform distribution on $\Delta^{M-1}$ in each iteration, however, the argument is abbreviated for the sake of simplicity. Then, the update rule with a gradient descent method described in Equation (11) is rewritten as

$$
\boldsymbol{X}^{(k)} = \boldsymbol{B}^{(k)} - \alpha^{(k)} \boldsymbol{G}^{(k)}.
$$

With this notation, the update rule for the control points in Equation (6) is represented as

$$
\boldsymbol{P}^{(k+1)} = \boldsymbol{P}^{(k)} - \alpha^{(k)} \left(\boldsymbol{Z}^{(k)\top} \boldsymbol{Z}^{(k)}\right)^{-1} \boldsymbol{Z}^{(k)\top} \boldsymbol{G}^{(k)}.
\tag{12}
$$

We summarize the pseudocode of the surface-wise gradient descent method in Algorithm 2.

## 6   PAC Stability of the Surface-wise Gradient Descent Method

We prove that the surface-wise gradient descent is PAC uniformly stable. All omitted proofs are shown in the Appendix. Hereinafter, we make the following assumption about the objective function.

**Assumption 6.1.** All the objective functions $f_1, \ldots, f_M$ are $\mu$-Lipschitz continuous and differentiable on $X$.

Let $\boldsymbol{x}^\star \colon \Delta^{M-1} \to X^\star(\boldsymbol{f})$ be a map from $\Delta^{M-1}$ to the Pareto set of $\boldsymbol{f}$. For $\boldsymbol{P}$ defined in Equation (2), we define a loss function as $\ell(\boldsymbol{P} \,|\, \boldsymbol{t}) := \|\boldsymbol{b}(\boldsymbol{t} \,|\, \boldsymbol{P}) - \boldsymbol{x}^\star(\boldsymbol{t})\|_2^2$. Since $X^*(\boldsymbol{f})$ is unknown, we cannot take a sample directly from $X^\star(\boldsymbol{f})$. Instead, we take a sample $\{\boldsymbol{t}_n\}_{n=1}^N$ drawn i.i.d. from the uniform distribution on $\Delta^{M-1}$.

To prove that Algorithm 2 is PAC uniformly stable, we show two propositions in advance. Note that the following two propositions can respectively be regarded as an extension of boundedness and expansiveness introduced in (Hardt et al., 2016) to analyze the stability of an optimization algorithm.

**Lemma 6.2.** *Let $U > 0$ be a constant satisfying $\max_{\boldsymbol{t} \in \Delta^{M-1}} \|\boldsymbol{z}(\boldsymbol{t})\|_2 \leq U$. Let $\varphi_{\boldsymbol{T}}$ be the update rule with parameter $\boldsymbol{T} = \{\boldsymbol{t}_n\}_{n=1}^N$ in Equation (12), i.e., $\boldsymbol{P}^{(k+1)} = \varphi_{\boldsymbol{T}}(\boldsymbol{P}^{(k)})$. Then there exists some $\eta > 0$, and we have the following inequality with probability at least $1 - \varepsilon$:*

$$\|\varphi_{\boldsymbol{T}}(\boldsymbol{P}) - \boldsymbol{P}\|_{\mathrm{F}} \leq \alpha^{(k)} \eta N U \mu.$$

**Lemma 6.3.** *Let $\eta > 0$ and $U > 0$ be constants as in Lemma 6.2. For $\boldsymbol{T} = \{\boldsymbol{t}_n\}_{n=1}^N$ and $\boldsymbol{T}' = \{\boldsymbol{t}_n'\}_{n=1}^N$ such that the difference between $\boldsymbol{T}$ and $\boldsymbol{T}'$ lies only in one example, there exists some $\zeta > 0$, and we have the following with probability at least $1 - \varepsilon$:*

$$\|\varphi_{\boldsymbol{T}}(\boldsymbol{P}) - \varphi_{\boldsymbol{T}'}(\boldsymbol{P})\|_{\mathrm{F}} \leq \mu U(\eta + \zeta N).$$

Let $\{\boldsymbol{T}_i\}_{i=1}^K$ and $\{\boldsymbol{T}_i'\}_{i=1}^K$ be parameters whose difference lies only in the $k$-th element, and $\boldsymbol{P}^{(K+1)}$ and $\boldsymbol{P}'^{(K+1)}$ be respectively the output of Algorithm 2 with $\{\boldsymbol{T}_i\}_{i=1}^K$ and $\{\boldsymbol{T}_i'\}_{i=1}^K$. From Lemmas 6.2 and 6.3, we show that $\left\|\boldsymbol{P}^{(K+1)} - \boldsymbol{P}'^{(K+1)}\right\|_{\mathrm{F}}$ is bounded above with arbitrary probability.

**Lemma 6.4.** *Let $U > 0$, $\eta > 0$ and $\zeta > 0$ be constants as in Lemma 6.3. Suppose that we run Algorithm 2 for $K$ iterations with parameters $\{\boldsymbol{T}_i\}_{i=1}^K$ and $\{\boldsymbol{T}_i'\}_{i=1}^K$ whose difference lies only in the kth element. Then, we have the following with probability at least $1 - \varepsilon$:*

$$\left\|\boldsymbol{P}^{(K+1)} - \boldsymbol{P}'^{(K+1)}\right\|_{\mathrm{F}} \leq 2\mu\eta U \left(1 + \left(K - k + \frac{\zeta}{\eta}\right)N\right).$$

Now, we are ready to show that Algorithm 2 is PAC uniformly stable.

**Theorem 6.5.** *Assume that $\alpha^{(k)} \in (0, 1]$ for all $k \in [K]$. Then, Algorithm 2 is PAC uniformly stable.*

From Theorems 4.2 and 6.5, if Algorithm 2 is decomposable, we can obtain an upper bound of its generalization gap.

*Remark* 6.6. While we assume the decomposability of Algorithm 2 in Theorem 6.5, this assumption is not restrictive in practice for the following reason. Suppose that an Algorithm 2 is not decomposable, meaning that there exists $\bar{\varepsilon} \in (0, 1)$ such that there are no event $B_{\bar{\varepsilon}}$ satisfying $D_{\bar{\varepsilon}} = B_{\bar{\varepsilon}}^{N+1}$. Even in this case, we can often take a smaller $\varepsilon' < \bar{\varepsilon}$ so that there exists an event $B_{\varepsilon'}$ satisfying $B_{\varepsilon'}^{N+1} \subseteq D_{\bar{\varepsilon}}$. Thus, by letting an event as $D_{\varepsilon'} \leftarrow B_{\varepsilon'}^{N+1}$, we can bound the difference between the generalization and empirical errors with an event $D_{\varepsilon'}$ which occurs with a certain probability.

# 7 Numerical Experiments

To verify that the Pareto optimal set can be accurately approximated by a Bézier simplex obtained by the proposed method (Algorithm 2), we applied Algorithm 2 to three multi-objective problems: scaled-MED, skew-$M$MED, and skew-$M$MMD employed in (Hamada et al., 2011), which are known to be simplicial. Skew-$M$MMD includes some important real-world problems such as a group LASSO in sparse modeling (Yuan & Lin, 2006), and skew-$M$MED includes a generalized location problem (Kuhn, 1967). For real-world instances, we applied Algorithm 2 to a group LASSO problem with the Birthwt dataset from the MASS package[1] in R language. In addition, to confirm the applicability of the proposed method to non-simplicial

---

[1] https://cran.r-project.org/web/packages/MASS/index.html

Table 1: MSE (avg.±s.d. over 100 trials) for scaled-MED.

| $N$ | Proposed (Algorithm 2) | NSGAII + all-at-once | MOEA/D + all-at-once |
|-----|------------------------|----------------------|----------------------|
| 30  | **6.07e-05** ± 3.18e-06 | 9.31e-02 ± 5.40e-04 | 3.32e-01 ± 4.28e-03 |
| 50  | **4.63e-05** ± 2.41e-06 | 1.37e-01 ± 8.18e-04 | 1.26e-01 ± 6.80e-04 |
| 100 | **3.83e-05** ± 1.59e-06 | 8.86e-02 ± 7.16e-04 | 1.14e-01 ± 6.62e-04 |

instances, we applied Algorithm 2 to DTLZ instances (Deb et al., 2002), which are widely used as benchmarks for multi-objective optimization algorithms. The definition of each problem instance is shown in Appendix E. In Algorithm 2, we set the degree of Bézier simplex as $D = 3$, the initial control points as $\boldsymbol{P}^{(1)} = \boldsymbol{O}$, which is the zero matrix of appropriate size. In addition, we set the maximum number of iterations as $K = 100$ and the step size as $\alpha^{(k)} = \frac{1}{k}$ for $k = 1, 2, \ldots, K$. The number of points to be sampled from a simplex in each iteration was tested for $N \in \{30, 50, 100\}$.

As a baseline, we used NSGA-II (Deb et al., 2000) and MOEA/D (Zhang & Li, 2007) with the Bézier simplex fitting (Kobayashi et al., 2019). Specifically, we obtained approximated Pareto solution samples by NSGA-II and MOEA/D implemented in jMetal (Benítez-Hidalgo et al., 2019), with default parameters except for population size. Then, we fitted the approximated Pareto solution samples with Bézier simplex of degree $D = 3$ by the all-at-once method proposed in (Kobayashi et al., 2019). We set the population size as $N \in \{30, 50, 100\}$. We implemented these algorithms in Python 3.12.3, and the experiments were performed on a Windows 10 PC with an Intel(R) Xeon(R) W-1270 CPU 3.40 GHz and 64 GB RAM. The source code is accessible at `https://github.com/hikimay/bezier-flow`.

## 7.1 MSEs comparison

First, we picked up a simplicial problem instance whose map $\boldsymbol{x}^\star \colon \Delta^{M-1} \to X^\star(\boldsymbol{f})$ is analytically obtained and evaluated how accurately an obtained Bézier simplex approximates the optimal Pareto set. In this experiment, we used scaled-MED, which is a three-objective problem with three variables and is known to be simplicial. The problem definition is shown in Appendix E. To evaluate the approximation accuracy of the estimated Bézier simplex, we used the mean squared error (MSE) defined by $\text{MSE} := \frac{1}{N} \sum_{n=1}^{N} \left\| \boldsymbol{b}(\hat{\boldsymbol{t}}_n \mid \boldsymbol{P}) - \boldsymbol{x}^\star(\hat{\boldsymbol{t}}_n) \right\|_2^2$, where $\boldsymbol{x}^\star$ is a map from a weight $\boldsymbol{t} \in \Delta^2$ to the minimizer $\boldsymbol{x}^\star(\boldsymbol{t})$ of the corresponding scalarizing function. The map $\boldsymbol{x}^\star$ for scaled-MED is shown in Appendix F. To calculate MSE, we randomly sample $\{\hat{\boldsymbol{t}}_n\}_{n=1}^{10000}$ i.i.d. from the uniform distribution on $\Delta^2$. We repeated the experiments 100 times with different parameters and computed the average and the standard deviations of MSEs.

Table 1 shows the average and the standard deviation of the MSEs with $N \in \{30, 50, 100\}$ for Algorithm 2 and population size $N \in \{30, 50, 100\}$ for NSGA-II and MOEA/D. In Table 1, we highlighted the best score of MSE out of the proposed and baseline methods where the difference is significant with the significance level p = 0.001 by the Wilcoxon rank-sum test. Table 1 implies that the Bézier simplex obtained by our proposed method can represent the Pareto set well. Moreover, the MSEs of our method decrease with larger $N$, which supports the PAC uniform stability of Algorithm 2.

Figure 3 shows the Bézier simplex obtained by our proposed method, and Figure 4 shows the Bézier simplex obtained by the all-at-once with the approximated Pareto solutions of NSGA-II. The true Pareto set of scaled-MED is known to be a curved triangle that can be triangulated into three vertices. Recall that the analytical solution of scaled-MED is shown in Appendix F. In Figure 3, the Bézier simplex obtained by the proposed method approximates the Pareto set well even when $N = 30$, while the Bézier simplex obtained by the all-at-once method with NSGA-II does not approximate the Pareto set even when $N = 100$. As a supplementary empirical evaluation, we conducted experiments to investigate the algorithmic stability of the proposed method on the scaled-MED. The details and results of this experiment are presented in Appendix H.

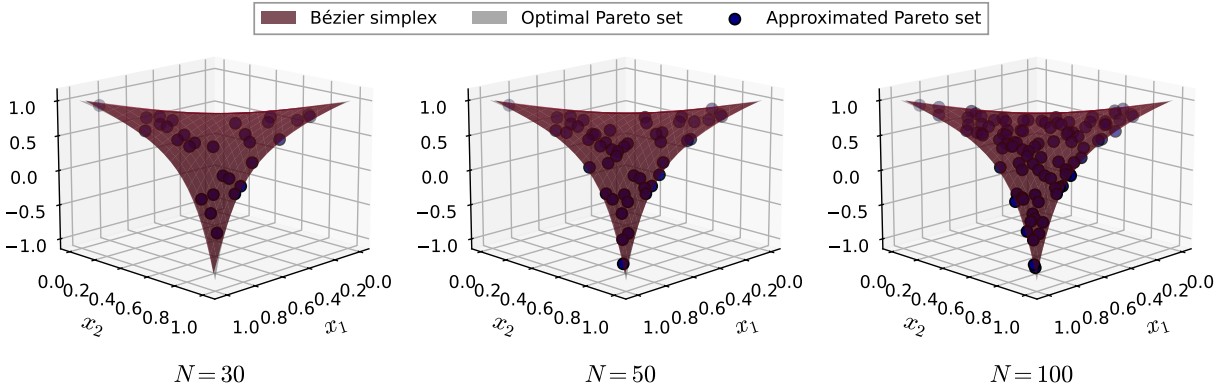

Figure 3: Results for Algorithm 2 with the sample size $N \in \{30, 50, 100\}$.

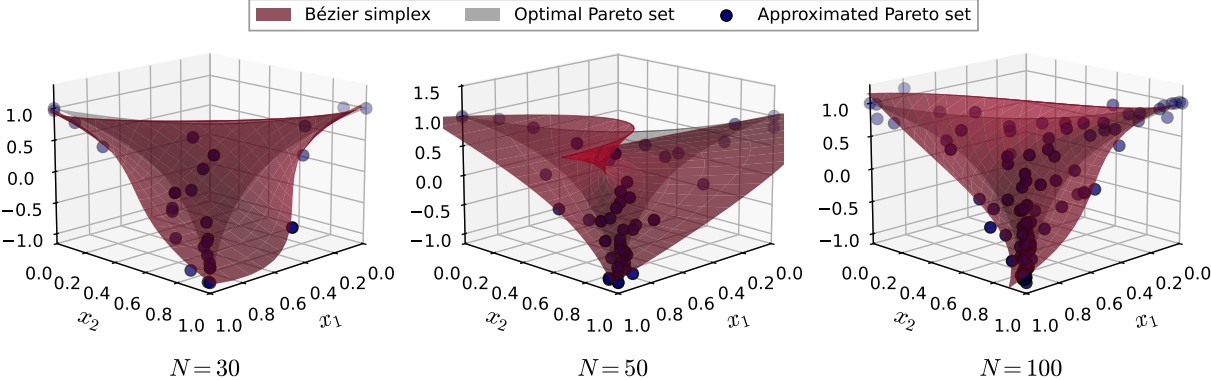

Figure 4: Results for NSGA-II and all-at-once with the population size $N \in \{30, 50, 100\}$.

## 7.2 GDs and IGDs comparison

Next, we validate the practicality of the proposed method in more practical settings. In this experiment, we used two synthetic simplicial problem instances (skew-3MED and skew-3MMD) and one real-world problem instance (group LASSO), whose map $\boldsymbol{x}^\star$ cannot be represented in a closed-form. Additionally, to confirm the applicability of the proposed method to non-simplicial instances, we applied Algorithm 2 to DTLZ instances (Deb et al., 2002) as non-simplicial instances. We show the definition of each problem instance in Appendix E. We used the generational distance (GD) (Veldhuizen, 1999) and inverted generational distance (IGD) (Zitzler et al., 2003) to evaluate how accurately the estimated Bézier simplex approximates the Pareto optimal set, which is defined as follows:

$$\mathrm{GD}(X, Y) := \frac{1}{|X|} \sum_{\boldsymbol{x} \in X} \min_{\boldsymbol{y} \in Y} \|\boldsymbol{x} - \boldsymbol{y}\|_2, \quad \mathrm{IGD}(X, Y) := \frac{1}{|Y|} \sum_{\boldsymbol{y} \in Y} \min_{\boldsymbol{x} \in X} \|\boldsymbol{x} - \boldsymbol{y}\|_2,$$

where $X$ is a finite set whose elements are sampled from an estimated hypersurface and $Y$ is a validation set. We can say that the obtained Bézier simplex is close to the Pareto set if and only if both GD and IGD are small. As a validation set $Y$, we generated approximate Pareto solutions by NSGA-III (Deb & Jain, 2013; Jain & Deb, 2013) with the population size of 1000. To construct $X$, we randomly sampled $\{\hat{\boldsymbol{t}}_n\}_{n=1}^{1000}$ i.i.d. from the uniform distribution on $\Delta^2$ and obtained sample points on the estimated Bézier simplex. We repeated the experiments 100 times with different parameters and computed the average and the standard deviations of their GDs and IGDs.

Tables 2 to 4 show the average and the standard deviation of the GDs and IGDs for skew-3MED, skew-3MMD, and group LASSO with the number of samples $N \in \{30, 50, 100\}$ and the population sizes $N \in \{30, 50, 100\}$.

The results for non-simplicial instances are shown in Appendix G. In Tables 2 to 4, we highlighted the best score of GD and IGD where the difference is at a significant with significance level p = 0.001 by the Wilcoxon rank-sum test. Tables 2 to 4 show that the proposed method achieved better GD and IGD for all cases. The differences are pronounced in the results of small sample/population size, which implies our method obtains a Bézier simplex approximating Pareto set well.

### 7.3 Run time comparison

To evaluate the computational efficiency, we measured the computation time of the proposed method (Algorithm 2) and two baseline methods (NSGAII and MOEA/D with all-at-once) for skew-3MED and skew-3MMD. In Table 5, we show the average and the standard deviation of the computation time in seconds of each method over ten trials. For the proposed method, we report the computation time with the maximum number of iterations $K = 100$. From Table 5, we see that the proposed method took about ten seconds to output the resulting Bezier simplex, whereas the post-processing method (NSGA-II and MOEA/D with Bezier simplex fitting) needed only about one to two seconds for each $N$. This is mainly because our method performs Bezier simplex fitting at every iteration, whereas existing methods do this only once. On the other hand, the computation time for the proposed methods remained almost the same even when $N$ increases by 100, while it grows for the existing methods. Although our method was slower in our experiments, its computing time did not increase much with larger $N$, which can be advantageous in certain situations.

## 8 Conclusion

This paper proposes a general framework to construct a multi-objective optimization algorithm from a single objective method with the Bézier simplex. We have also defined the PAC stability of optimization algorithms and proved that this stability gives us an upper bound on the generalization gap in the sense of PAC learning. The theoretical analysis showed that if we construct a multi-objective optimization algorithm from a gradient descent-based single-objective optimization algorithm, the resultant algorithm is PAC stable. In our numerical experiments, we applied the multi-objective optimization algorithm with our framework to three synthetic and one real-world instances and demonstrated that our algorithm attained better generalization gaps and approximation accuracies of the Pareto optimal set than the existing algorithm. As a concluding remark, we have to note that this study is limited to treat simplicial problems. It would be interesting for future studies to extend this study to non-simplicial cases.

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

Table 2: GD and IGD (avg.±s.d. over 100 trials) for skew-3MED.

|  | $N$ | Proposed (Algorithm 2) | NSGAII + all-at-once | MOEA/D + all-at-once |
|---|---|---|---|---|
| GD | 30 | **7.07e-02** $\pm$ 2.05e-03 | 2.59e-01 $\pm$ 7.41e-03 | 1.57e-01 $\pm$ 5.42e-03 |
|  | 50 | **6.53e-02** $\pm$ 1.18e-03 | 1.26e-01 $\pm$ 4.72e-03 | 9.36e-02 $\pm$ 1.46e-03 |
|  | 100 | **6.59e-02** $\pm$ 1.23e-04 | 7.57e-02 $\pm$ 1.11e-03 | 6.63e-02 $\pm$ 9.19e-04 |
| IGD | 30 | **9.91e-02** $\pm$ 8.43e-04 | 1.69e-01 $\pm$ 5.39e-03 | 1.28e-01 $\pm$ 3.35e-03 |
|  | 50 | **9.75e-02** $\pm$ 4.84e-04 | 1.15e-01 $\pm$ 1.89e-02 | 1.09e-01 $\pm$ 3.47e-03 |
|  | 100 | **8.98e-02** $\pm$ 5.22e-04 | 9.19e-02 $\pm$ 3.44e-03 | 9.77e-02 $\pm$ 2.78e-03 |

Table 3: GD and IGD (avg.±s.d. over 100 trials) for skew-3MMD.

|  | $N$ | Proposed (Algorithm 2) | NSGAII + all-at-once | MOEA/D + all-at-once |
|---|---|---|---|---|
| GD | 30 | **5.57e-02** $\pm$ 1.69e-03 | 1.61e-01 $\pm$ 3.19e-03 | 5.94e-02 $\pm$ 9.98e-04 |
|  | 50 | **5.33e-02** $\pm$ 1.25e-03 | 9.98e-02 $\pm$ 1.98e-03 | 5.94e-02 $\pm$ 9.86e-04 |
|  | 100 | **5.14e-02** $\pm$ 1.02e-03 | 8.57e-02 $\pm$ 2.50e-02 | 5.94e-02 $\pm$ 2.54e-03 |
| IGD | 30 | **6.69e-02** $\pm$ 6.73e-04 | 8.57e-02 $\pm$ 2.78e-03 | 8.58e-02 $\pm$ 3.16e-03 |
|  | 50 | **6.34e-02** $\pm$ 5.57e-04 | 7.17e-02 $\pm$ 2.48e-03 | 6.45e-02 $\pm$ 3.06e-03 |
|  | 100 | **6.28e-02** $\pm$ 4.62e-04 | 1.01e-01 $\pm$ 6.31e-02 | 7.31e-02 $\pm$ 2.95e-03 |

Table 4: GD and IGD (avg.±s.d. over 100 trials) for group LASSO.

|  | $N$ | Proposed (Algorithm 2) | NSGAII + all-at-once | MOEA/D + all-at-once |
|---|---|---|---|---|
| GD | 30 | **2.44e-02** $\pm$ 5.25e-04 | 6.47e-02 $\pm$ 1.75e-04 | 6.78e-02 $\pm$ 2.78e-03 |
|  | 50 | **2.37e-02** $\pm$ 4.72e-04 | 4.29e-02 $\pm$ 3.73e-04 | 4.13e-02 $\pm$ 4.51e-04 |
|  | 100 | **2.37e-02** $\pm$ 4.30e-04 | 4.56e-02 $\pm$ 5.06e-04 | 4.12e-02 $\pm$ 3.10e-04 |
| IGD | 30 | **5.94e-02** $\pm$ 3.98e-04 | 2.05e-01 $\pm$ 4.12e-04 | 1.35e-01 $\pm$ 2.76e-04 |
|  | 50 | **5.95e-02** $\pm$ 3.79e-04 | 1.18e-01 $\pm$ 3.99e-04 | 9.96e-02 $\pm$ 3.97e-04 |
|  | 100 | **5.92e-02** $\pm$ 3.82e-04 | 1.24e-01 $\pm$ 5.13e-04 | 6.44e-02 $\pm$ 3.12e-04 |

Table 5: CPU time in seconds for skew-3MED and skew-3MMD.

|  | $N$ | Proposed (Algorithm 2) | NSGAII + all-at-once | MOEA/D + all-at-once |
|---|---|---|---|---|
| skew-3MED | 30 | 1.09e+01 $\pm$ 5.79e-01 | 1.41e-00 $\pm$ 1.09e-01 | 1.92e+00 $\pm$ 4.92e-02 |
|  | 50 | 1.07e+01 $\pm$ 7.25e-01 | 1.67e-00 $\pm$ 4.06e-02 | 2.04e+00 $\pm$ 5.38e-02 |
|  | 100 | 1.08e+01 $\pm$ 6.72e-01 | 2.57e-00 $\pm$ 8.76e-02 | 1.85e+00 $\pm$ 3.74e-02 |
| skew-3MMD | 30 | 1.32e+01 $\pm$ 7.26e-01 | 1.39e-00 $\pm$ 8.89e-03 | 2.04e+00 $\pm$ 5.38e-02 |
|  | 50 | 1.29e+01 $\pm$ 5.84e-01 | 1.83e-00 $\pm$ 9.01e-02 | 2.12e+00 $\pm$ 2.34e-02 |
|  | 100 | 1.30e+01 $\pm$ 6.95e-01 | 2.69e-00 $\pm$ 7.59e-02 | 2.43e+00 $\pm$ 2.89e-02 |

Kalyanmoy Deb, Samir Agrawal, Amrit Pratap, and T. Meyarivan. A fast and elitist multiobjective genetic algorithm: NSGA-II. *IEEE Transactions on Evolutionary Computation*, 6(2):182–197, April 2002. doi: 10.1109/4235.996017.

Gabriele Eichfelder. *Adaptive Scalarization Methods in Multiobjective Optimization*. Springer-Verlag, Berlin, Heidelberg, 2008.

Naoki Hamada and Shunsuke Ichiki. Simpliciality of strongly convex problems. *Journal of the Mathematical Society of Japan*, 73(3), 2020. doi: 10.2969/jmsj/83918391.

Naoki Hamada, Yuichi Nagata, Shigenobu Kobayashi, and Isao Ono. On scalability of adaptive weighted aggregation for multiobjective function optimization. In *2011 IEEE Congress of Evolutionary Computation (CEC)*, pp. 669–678, 2011. doi: 10.1109/CEC.2011.5949683.

Naoki Hamada, Kenta Hayano, Shunsuke Ichiki, Yutaro Kabata, and Hiroshi Teramoto. Topology of Pareto sets of strongly convex problems. *SIAM Journal on Optimization*, 30(3):2659–2686, 2020. doi: 10.1137/19M1271439.

Ken Harada, Jun Sakuma, Shigenobu Kobayashi, and Isao Ono. Uniform sampling of local Pareto-optimal solution curves by Pareto path following and its applications in multi-objective GA. In *Proceedings of the Genetic and Evolutionary Computation Conference 2007*, pp. 813–820. ACM, 2007. doi: 10.1145/1276958. 1277120.

Moritz Hardt, Ben Recht, and Yoram Singer. Train faster, generalize better: Stability of stochastic gradient descent. In *Proceedings of the 33rd International Conference on Machine Learning*, volume 48, pp. 1225–1234. PMLR, 2016.

Daniel Hernandez-Lobato, Jose Hernandez-Lobato, Amar Shah, and Ryan Adams. Predictive entropy search for multi-objective bayesian optimization. In *Proceedings of the 33rd International Conference on Machine Learning*, volume 48, pp. 1492–1501. PMLR, 2016.

Claus Hillermeier. *Nonlinear Multiobjective Optimization: A Generalized Homotopy Approach*, volume 25 of *International Series of Numerical Mathematics*. Birkhäuser Verlag, Basel, Boston, Berlin, 2001. doi: 10.1007/978-3-0348-8280-4.

Himanshu Jain and Kalyanmoy Deb. An evolutionary many-objective optimization algorithm using reference-point based nondominated sorting approach, Part II: Handling constraints and extending to an adaptive approach. *IEEE Transactions on Evolutionary Computation*, 18(4):602–622, 2013.

Ken Kobayashi, Naoki Hamada, Akiyoshi Sannai, Akinori Tanaka, Kenichi Bannai, and Masashi Sugiyama. Bézier simplex fitting: Describing Pareto fronts of simplicial problems with small samples in multi-objective optimization. In *Proceedings of the AAAI Conference on Artificial Intelligence*, volume 33, pp. 2304–2313, 2019. doi: 10.1609/aaai.v33i01.33012304.

Harold W. Kuhn. On a pair of dual nonlinear programs. *Nonlinear Programming*, 1:38–45, 1967.

Stefanus C. Maree, Tanja Alderliesten, and Peter A. N. Bosman. Ensuring smoothly navigable approximation sets by bézier curve parameterizations in evolutionary bi-objective optimization. In *Parallel Problem Solving from Nature – PPSN XVI*, pp. 215–228. 2020. URL https://doi.org/10.1007/978-3-030-58115-2_15.

Franco Mastroddi and Stefania Gemma. Analysis of Pareto frontiers for multidisciplinary design optimization of aircraft. *Aerospace Science and Technology*, 28(1):40–55, 2013. doi: 10.1016/j.ast.2012.10.003.

Kaisa Miettinen. *Nonlinear Multiobjective Optimization*. Springer US, 1999. doi: 10.1007/978-1-4615-5563-6.

Yusuke Mizota, Naoki Hamada, and Shunsuke Ichiki. All unconstrained strongly convex problems are weakly simplicial. *arXiv,* arXiv:2106.12704, 2021.

Gilberto Reynoso-Meza, Leandro dos Santos Coelho, and Roberto Z. Freite. Efficient sampling of pi controllers in evolutionary multiobjective optimization. In *Proceedings of the 2015 Annual Conference on Genetic and Evolutionary Computation*, pp. 1263–1270, New York, NY, USA, 2015. Association for Computing Machinery. ISBN 9781450334723. doi: 10.1145/2739480.2754807.

Siamak Safarzadegan Gilan, Naman Goyal, and Bistra Dilkina. Active learning in multi-objective evolutionary algorithms for sustainable building design. In *Proceedings of the Genetic and Evolutionary Computation Conference 2016*, pp. 589–596, 2016. doi: 10.1145/2908812.2908947.

Oren Shoval, Hila Sheftel, Guy Shinar, Yuval Hart, Omer Ramote, Avi Mayo, Erez Dekel, Kavanagh Kavanagh, and Uri Alon. Evolutionary trade-offs, Pareto optimality, and the geometry of phenotype space. *Science*, 336(6085):1157–1160, 2012. doi: 10.1126/science.1217405.

Akinori Tanaka, Akiyoshi Sannai, Ken Kobayashi, and Naoki Hamada. Asymptotic risk of bézier simplex fitting. In *Proceedings of the AAAI Conference on Artificial Intelligence*, pp. 2416–2424, 2020. doi: 10.1609/aaai.v34i03.5622.

David Allen Van Veldhuizen. *Multiobjective Evolutionary Algorithms: Classifications, Analyses, and New Innovations.* PhD thesis, Department of Electrical and Computer Engineering, Graduate School of Engineering, Air Force Institute of Technology, 1999.

Jasper A. Vrugt, Hoshin V. Gupta, Luis A. Bastidas, Willem Bouten, and Soroosh Sorooshian. Effective and efficient algorithm for multiobjective optimization of hydrologic models. *Water Resources Research*, 39(8): 1214–1232, 2003. doi: 10.1029/2002WR001746.

Yieh-Hei Wan. On the algebraic criteria for local Pareto optima–I. *Topology*, 16:113–117, 1977. doi: 10.1016/0040-9383(77)90035-0.

Yieh-Hei Wan. On the algebraic criteria for local Pareto optima. II. *Transactions of the American Mathematical Society*, 245:385–397, 1978.

Kaifeng Yang, Michael Emmerich, André Deutz, and Thomas Bäck. Multi-objective Bayesian global optimization using expected hypervolume improvement gradient. *Swarm and Evolutionary Computation*, 44:945–956, 2019. doi: 10.1016/j.swevo.2018.10.007.

Ming Yuan and Yi Lin. Model selection and estimation in regression with grouped variables. *Journal of the Royal Statistical Society: Series B (Statistical Methodology)*, 68(1):49–67, 2006.

Qingfu Zhang and Hui Li. MOEA/D: A multiobjective evolutionary algorithm based on decomposition. *IEEE Transactions on Evolutionary Computation*, 11(6):712–731, 2007. doi: 10.1109/TEVC.2007.892759.

Eckart Zitzler, Lothar Thiele, Marco Laumanns, Carlos M. Fonseca, and Vviane Grunert Da Fonseca. Performance assessment of multiobjective optimizers: An analysis and review. *IEEE Transactions on Evolutionary Computation*, 7(2):117–132, 2003. doi: 10.1109/TEVC.2003.810758.

## A   Proof of Theorem 4.2

*Proof.* We denote two independent random samples by $S = (\boldsymbol{t}_1, \ldots, \boldsymbol{t}_N)$, $S' = (\boldsymbol{t}'_1, \ldots, \boldsymbol{t}'_N)$. Let $S^{(i)} = (\boldsymbol{t}_1, \ldots, \boldsymbol{t}_{i-1}, \boldsymbol{t}'_i, \boldsymbol{t}_{i+1}, \ldots, \boldsymbol{t}_N)$ be the sample that is same as $S$ except in the $i$th example where we replace $\boldsymbol{t}_i$ with $\boldsymbol{t}'_i$. Since $D_\varepsilon = B_\varepsilon^{N+1}$, $(\boldsymbol{t}_1, \ldots, \boldsymbol{t}_i, \boldsymbol{t}'_i, \boldsymbol{t}_{i+1}, \ldots, \boldsymbol{t}_N) \in D_\varepsilon$ for any $i$ if and only if $S, S' \in C_\varepsilon := B_\varepsilon^N$. In this case, we have

$$\mathbb{E}_{\mathcal{A}}\left[\left|\ell(\mathcal{A}(S) \,|\, \boldsymbol{t}_i) - \ell(\mathcal{A}(S^{(i)}) \,|\, \boldsymbol{t}_i)\right|\right] < \delta.$$

Then, adding the inequalities for $i$ and applying the triangle inequality, we obtain

$$\left|\mathbb{E}_{\mathcal{A}}\left[\frac{1}{N}\sum_{i=1}^{N}\ell(\mathcal{A}(S) \,|\, \boldsymbol{t}_i) - \frac{1}{N}\sum_{i=1}^{N}\ell(\mathcal{A}(S^{(i)}) \,|\, \boldsymbol{t}_i)\right]\right| < \delta.$$

Let us denote the conditional probability distribution of $\mathcal{D}, \mathcal{D}^N$ under the condition $B_\varepsilon, C_\varepsilon$ by $\mathcal{B}_\varepsilon, \mathcal{C}_\varepsilon$ respectively. Then we have

$$\left|\mathbb{E}_{(S,S')\sim\mathcal{C}_\varepsilon^2}\mathbb{E}_{\mathcal{A}}\left[\frac{1}{N}\sum_{i=1}^{N}\ell(\mathcal{A}(S) \,|\, \boldsymbol{t}_i) - \frac{1}{N}\sum_{i=1}^{N}\ell(\mathcal{A}(S^{(i)}) \,|\, \boldsymbol{t}_i)\right]\right| < \delta.$$

Here, we have

$$\mathbb{E}_{(S,S')\sim\mathcal{C}_\varepsilon^2}\mathbb{E}_{\mathcal{A}}\left[\frac{1}{N}\sum_{i=1}^{N}\ell(\mathcal{A}(S) \,|\, \boldsymbol{t}_i)\right] = \mathbb{E}_{S\sim\mathcal{C}_\varepsilon}\mathbb{E}_{\mathcal{A}}\left[\frac{1}{N}\sum_{i=1}^{N}\ell(\mathcal{A}(S) \,|\, \boldsymbol{t}_i)\right] = \hat{\mathbb{E}}_S\mathbb{E}_{\mathcal{A}}[R_S[\mathcal{A}(S)]],$$

and

$$\mathbb{E}_{(S,S')\sim\mathcal{C}_\varepsilon^2}\mathbb{E}_\mathcal{A}\left[\frac{1}{N}\sum_{i=1}^N \ell(\mathcal{A}(S^{(i)})\,|\,\boldsymbol{t}_i)\right] = \mathbb{E}_{(S,S')\sim\mathcal{C}_\varepsilon^2}\mathbb{E}_\mathcal{A}\left[\frac{1}{N}\sum_{i=1}^N \ell(\mathcal{A}(S)\,|\,\boldsymbol{t}_i')\right]$$

$$= \mathbb{E}_{S\sim\mathcal{C}_\varepsilon}\mathbb{E}_\mathcal{A}\mathbb{E}_{S'\sim\mathcal{C}_\varepsilon}\left[\frac{1}{N}\sum_{i=1}^N \ell(\mathcal{A}(S)\,|\,\boldsymbol{t}_i')\right]$$

$$= \hat{\mathbb{E}}_S\mathbb{E}_\mathcal{A}\left[\hat{R}[\mathcal{A}(S)]\right],$$

where $\hat{\mathbb{E}}_S$ is the conditional expected value of $C_\varepsilon$ and $\hat{R}[\mathcal{A}(S)]$ is the conditional generalization error of $C_\varepsilon$. Thus, we obtain the inequality in the theorem.

Finally, we have

$$\mathbb{P}(C_\varepsilon) = \mathbb{P}(B_\varepsilon)^N = \mathbb{P}(D_\varepsilon)^{\frac{N}{N+1}} > \mathbb{P}(D_\varepsilon) > 1 - \varepsilon,$$

which completes the proof. □

## B    Proofs of Lemmas 6.2 and 6.3

We first show the three lemmas in advance.

**Lemma B.1.** *For all $\varepsilon \in (0,1)$, there exists $\eta > 0$ satisfying*

$$\mathbb{P}\left(\min_i \lambda_{\min}\left(\boldsymbol{Z}(\boldsymbol{T}_i)^\top \boldsymbol{Z}(\boldsymbol{T}_i)\right) > \eta\right) \geq 1 - \varepsilon,$$

*where $\lambda_{\min}(\boldsymbol{A})$ denotes the minimal eigenvalue of a symmetric matrix $\boldsymbol{A}$, and $\boldsymbol{T}_i = \{\boldsymbol{t}_n^{(i)}\}_{n=1}^N$ is drawn i.i.d. from the uniform distribution on $\Delta^{M-1}$ for $i \in [K+1]$.*

*Proof.* Consider the set

$$\mathcal{F}_0 := \left\{\{\boldsymbol{T_i}\}_{i=1}^{K+1} \subseteq \left(\Delta^{M-1}\right)^{K+1} \,\middle|\, \min_i \lambda_{\min}\left(\boldsymbol{Z}(\boldsymbol{T_i})^\top \boldsymbol{Z}(\boldsymbol{T_i})\right) = 0\right\}.$$

Since $\boldsymbol{Z}(\boldsymbol{T}_i)^\top \boldsymbol{Z}(\boldsymbol{T}_i)$ is a symmetric matrix, it has non-negative eigenvalues and $\boldsymbol{Z}(\boldsymbol{T}_i)^\top \boldsymbol{Z}(\boldsymbol{T}_i)$ has zero eigenvalue if and only if the determinant of $\boldsymbol{Z}(\boldsymbol{T}_i)^\top \boldsymbol{Z}(\boldsymbol{T}_i)$ is zero. Hence, $\mathcal{F}_0$ is a subset of

$$\left\{\{\boldsymbol{t}_n\}_{n=1}^N \subseteq \Delta^{M-1} \,\middle|\, \prod_i \det\left(\boldsymbol{Z}(\boldsymbol{T}_i)^\top \boldsymbol{Z}(\boldsymbol{T}_i)\right) = 0\right\}.$$

Therefore, $\mathcal{F}_0$ is equal to the zero set of a polynomial. This implies $\mathbb{P}(\mathcal{F}_0) = 0$. Considering the set

$$\mathcal{F}_\eta := \left\{\{\boldsymbol{T_i}\}_{i=1}^{K+1} \subseteq \left(\Delta^{M-1}\right)^{K+1} \,\middle|\, \min_i \lambda_{\min}\left(\boldsymbol{Z}(\boldsymbol{T_i})^\top \boldsymbol{Z}(\boldsymbol{T_i})\right) \leq \eta\right\},$$

then we have

$$\mathbb{P}(\mathcal{F}_\eta) \to \mathbb{P}(\mathcal{F}_0) = 0 \quad (\eta \to 0).$$

This implies that for all $\varepsilon \in (0,1)$ there exists some $\eta > 0$ such that

$$\mathbb{P}\left(\min_i \lambda_{\min}\left(\boldsymbol{Z}(\boldsymbol{T}_i)^\top \boldsymbol{Z}(\boldsymbol{T}_i)\right) \leq \eta\right) < \varepsilon, \tag{13}$$

The proof is completed by taking the complementary event. □

**Lemma B.2.** *For all $\varepsilon \in (0,1)$, there exists $\zeta > 0$ satisfying*

$$\mathbb{P}\left(\max_i \left\|\boldsymbol{Z}(\boldsymbol{T}_i)^\top \boldsymbol{Z}(\boldsymbol{T}_i)\right\|_{\mathrm{F}} < \zeta\right) \geq 1 - \varepsilon, \tag{14}$$

*where $\boldsymbol{T}_i = \{\boldsymbol{t}_n^{(i)}\}_{n=1}^N$ is drawn i.i.d. from the uniform distribution on $\Delta^{M-1}$ for $i \in [K+1]$.*

*Proof.* Since $\boldsymbol{Z}(\boldsymbol{T}_i)^\top \boldsymbol{Z}(\boldsymbol{T}_i)$ is a symmetric matrix, there exists an orthogonal matrix $\boldsymbol{Q}_i$ such that $\boldsymbol{Z}(\boldsymbol{T}_i)^\top \boldsymbol{Z}(\boldsymbol{T}_i) = \boldsymbol{Q}_i \boldsymbol{\Lambda}_i \boldsymbol{Q}_i^\top$ where $\boldsymbol{\Lambda}_i$ is a diagonal matrix whose diagonal entry is the eigenvalue of $\boldsymbol{Z}(\boldsymbol{T}_i)^\top \boldsymbol{Z}(\boldsymbol{T}_i)$. According to Lemma B.1, $\boldsymbol{Z}(\boldsymbol{T}_i)^\top \boldsymbol{Z}(\boldsymbol{T}_i)$ is a regular matrix with probability at least $1 - \varepsilon$ for all $i \in [K+1]$. Hence, we have the following with probability at least $1 - \varepsilon$:

$$
\begin{aligned}
\max_i \left\| \left( \boldsymbol{Z}(\boldsymbol{T}_i)^\top \boldsymbol{Z}(\boldsymbol{T}_i) \right)^{-1} \right\|_{\mathrm{F}} &= \max_i \left\| \boldsymbol{Q}_i^\top \boldsymbol{\Lambda}_i^{-1} \boldsymbol{Q}_i \right\|_{\mathrm{F}} \\
&= \max_i \left\| \boldsymbol{\Lambda}_i^{-1} \right\|_{\mathrm{F}} \\
&= \max_i \sqrt{ \sum_{n=1}^{|\mathbb{N}_D^M|} \frac{1}{\lambda_n^2 \left( \boldsymbol{Z}(\boldsymbol{T}_i)^\top \boldsymbol{Z}(\boldsymbol{T}_i) \right)} } \\
&\leq \sqrt{ \frac{|\mathbb{N}_D^M|}{\min_i \lambda_{\min}^2 \left( \boldsymbol{Z}(\boldsymbol{T}_i)^\top \boldsymbol{Z}(\boldsymbol{T}_i) \right)} } \\
&< \frac{\sqrt{|\mathbb{N}_D^M|}}{\eta},
\end{aligned}
$$

where the second equality follows from the fact that the Frobenius norm is unitarily invariant, and the second inequality follows from Lemma B.1. The proof is completed by letting $\zeta$ be some real number greater than or equal to $\frac{\sqrt{|\mathbb{N}_D^M|}}{\eta}$. $\qquad\square$

**Lemma B.3.** *Let $U > 0$ be a constant satisfying $\max_{\boldsymbol{t} \in \Delta^{M-1}} \|\boldsymbol{z}(\boldsymbol{t})\|_2 \leq U$. Then, for any $\boldsymbol{T} \subseteq \Delta^{M-1}$, we have*

$$
\left\| \boldsymbol{Z}(\boldsymbol{T})^\top \boldsymbol{G}(\boldsymbol{T}) \right\|_{\mathrm{F}} \leq N U \mu. \tag{15}
$$

*Proof.* First, we show that $\|\boldsymbol{z}(\boldsymbol{t})\|_2$ is bounded above for any $\boldsymbol{t} \in \Delta^{M-1}$. Since $\boldsymbol{z}$ is a continuous function over $\boldsymbol{t}$ whose domain $\Delta^{M-1}$ is compact, there exists upper bound $U > 0$ for any $\boldsymbol{t} \in \Delta^{M-1}$. Next, we show that $\left\| \boldsymbol{Z}(\boldsymbol{T})^\top \boldsymbol{G}(\boldsymbol{T}) \right\|_{\mathrm{F}}$ is bounded above. For any $\boldsymbol{T} := \{\boldsymbol{t}_n\}_{n=1}^N \subseteq \Delta^{M-1}$, we have

$$
\boldsymbol{Z}(\boldsymbol{T})^\top \boldsymbol{G}(\boldsymbol{T}) = (\boldsymbol{z}_1, \ldots, \boldsymbol{z}_N) \begin{bmatrix} \boldsymbol{t}_1^\top J_{\boldsymbol{f}} (\boldsymbol{P}^\top \boldsymbol{z}_1) \\ \vdots \\ \boldsymbol{t}_N^\top J_{\boldsymbol{f}} (\boldsymbol{P}^\top \boldsymbol{z}_N) \end{bmatrix} = \sum_{n=1}^N \boldsymbol{z}_n \boldsymbol{t}_n^\top J_{\boldsymbol{f}} (\boldsymbol{P}^\top \boldsymbol{z}_n).
$$

Therefore,

$$
\begin{aligned}
\left\| \boldsymbol{Z}(\boldsymbol{T})^\top \boldsymbol{G}(\boldsymbol{T}) \right\|_{\mathrm{F}} &= \left\| \sum_{n=1}^N \boldsymbol{z}_n \boldsymbol{t}_n^\top J_{\boldsymbol{f}} (\boldsymbol{P}^\top \boldsymbol{z}_n) \right\|_{\mathrm{F}} \\
&\leq \sum_{n=1}^N \left\| \boldsymbol{z}_n \boldsymbol{t}_n^\top J_{\boldsymbol{f}} (\boldsymbol{P}^\top \boldsymbol{z}_n) \right\|_{\mathrm{F}} \\
&\leq \sum_{n=1}^N \|\boldsymbol{z}_n\|_2 \cdot \left\| \boldsymbol{t}_n^\top J_{\boldsymbol{f}} (\boldsymbol{P}^\top \boldsymbol{z}_n) \right\|_2 \\
&= \sum_{n=1}^N \|\boldsymbol{z}_n\|_2 \cdot \left\| \sum_{m=1}^M t_{nm} \nabla f_m (\boldsymbol{P}^\top \boldsymbol{z}) \right\|_2 \\
&\leq \sum_{n=1}^N U \mu = N U \mu.
\end{aligned}
$$

The last inequality holds by the fact that the term $\sum_{m=1}^{M} t_{nm} \nabla f_m(\boldsymbol{P}^\top \boldsymbol{z})$ is a convex combination of $\nabla f_1(\boldsymbol{P}^\top \boldsymbol{z}), \ldots, \nabla f_M(\boldsymbol{P}^\top \boldsymbol{z})$ and the assumption that every function $f_1, \ldots, f_M$ is $\mu$-Lipschitz continuous. $\qquad\square$

Finally, we show Lemmas 6.2 and 6.3.

*Proof of Lemma 6.2.* By Equation (12), we have

$$\begin{aligned}
\|\varphi_{\boldsymbol{T}}(\boldsymbol{P}) - \boldsymbol{P}\|_{\mathrm{F}} &= \alpha^{(k)} \left\| \left(\boldsymbol{Z}^\top \boldsymbol{Z}\right)^{-1} \boldsymbol{Z}^\top \boldsymbol{G} \right\|_{\mathrm{F}} \\
&\le \alpha^{(k)} \left\| \left(\boldsymbol{Z}^\top \boldsymbol{Z}\right)^{-1} \right\|_{\mathrm{F}} \cdot \left\| \boldsymbol{Z}^\top \boldsymbol{G} \right\|_{\mathrm{F}}.
\end{aligned}$$

Let $\eta > 0$ be a constant as in Lemma B.2. From Lemmas B.2 and B.3, we have the following with probability at least $1 - \varepsilon$:

$$\|\varphi_{\boldsymbol{t}}(\boldsymbol{P}) - \boldsymbol{P}\|_{\mathrm{F}} \le \alpha^{(k)} \eta N U \mu.$$

$\qquad\square$

*Proof of Lemma 6.3.* Let $\boldsymbol{T}$, $\boldsymbol{T}'$ and $\widetilde{\boldsymbol{T}}$ be

$$\begin{aligned}
\boldsymbol{T} &= \{\boldsymbol{t}_1, \ldots, \boldsymbol{t}_{N-1}, \boldsymbol{t}_N\}, \\
\boldsymbol{T}' &= \{\boldsymbol{t}_1, \ldots, \boldsymbol{t}_{N-1}, \boldsymbol{t}'_N\}, \\
\widetilde{\boldsymbol{T}} &= \{\boldsymbol{t}_1, \ldots, \boldsymbol{t}_{N-1}, \boldsymbol{t}_N, \boldsymbol{t}'_N\}.
\end{aligned}$$

Let $\widetilde{\boldsymbol{Z}}$ be a matrix constructed by $\widetilde{\boldsymbol{T}}$. By Sherman-Morrison formula, we have

$$\begin{aligned}
\left(\widetilde{\boldsymbol{Z}}^\top \widetilde{\boldsymbol{Z}}\right)^{-1} &= \left(\boldsymbol{Z}^\top \boldsymbol{Z} + \boldsymbol{z}_{N+1} \boldsymbol{z}_{N+1}^\top\right)^{-1} \\
&= \left(\boldsymbol{Z}^\top \boldsymbol{Z}\right)^{-1} + \frac{\left(\boldsymbol{Z}^\top \boldsymbol{Z}\right)^{-1} \boldsymbol{z}_{N+1} \boldsymbol{z}_{N+1}^\top \left(\boldsymbol{Z}^\top \boldsymbol{Z}\right)^{-1}}{1 + \boldsymbol{z}_{N+1}^\top \left(\boldsymbol{Z}^\top \boldsymbol{Z}\right)^{-1} \boldsymbol{z}_{N+1}}.
\end{aligned}$$

Let $\varphi_{\boldsymbol{T}}(\boldsymbol{P})$ be the control points obtained by Algorithm 2 with $\boldsymbol{T}$. Then, we have

$$\begin{aligned}
\varphi_{\widetilde{\boldsymbol{T}}}(\boldsymbol{P}) - \varphi_{\boldsymbol{T}}(\boldsymbol{P}) &= \left(\widetilde{\boldsymbol{Z}}^\top \widetilde{\boldsymbol{Z}}\right)^{-1} \widetilde{\boldsymbol{Z}}^\top \widetilde{\boldsymbol{G}} - \left(\boldsymbol{Z}^\top \boldsymbol{Z}\right)^{-1} \boldsymbol{Z}^\top \boldsymbol{G} \\
&= \left(\boldsymbol{Z}^\top \boldsymbol{Z}\right)^{-1} \left(\widetilde{\boldsymbol{Z}}^\top \widetilde{\boldsymbol{G}} - \boldsymbol{Z}^\top \boldsymbol{G}\right) + \frac{\left(\boldsymbol{Z}^\top \boldsymbol{Z}\right)^{-1} \boldsymbol{z}_{N+1} \boldsymbol{z}_{N+1}^\top \left(\boldsymbol{Z}^\top \boldsymbol{Z}\right)^{-1} \widetilde{\boldsymbol{Z}}^\top \widetilde{\boldsymbol{G}}}{1 + \boldsymbol{z}_{N+1}^\top \left(\boldsymbol{Z}^\top \boldsymbol{Z}\right)^{-1} \boldsymbol{z}_{N+1}} \\
&= \left(\boldsymbol{Z}^\top \boldsymbol{Z}\right)^{-1} \left(\boldsymbol{z}_{N+1} \boldsymbol{t}_{N+1}^\top J_{\boldsymbol{f}}\left(\boldsymbol{P}^\top \boldsymbol{z}_{N+1}\right)\right) + \frac{\left(\boldsymbol{Z}^\top \boldsymbol{Z}\right)^{-1} \boldsymbol{z}_{N+1} \boldsymbol{z}_{N+1}^\top \left(\boldsymbol{Z}^\top \boldsymbol{Z}\right)^{-1} \widetilde{\boldsymbol{Z}}^\top \widetilde{\boldsymbol{G}}}{1 + \boldsymbol{z}_{N+1}^\top \left(\boldsymbol{Z}^\top \boldsymbol{Z}\right)^{-1} \boldsymbol{z}_{N+1}}.
\end{aligned}$$

Considering the norm on both sides, we have

$$\begin{aligned}
\left\| \varphi_{\widetilde{\boldsymbol{T}}}(\boldsymbol{P}) - \varphi_{\boldsymbol{T}}(\boldsymbol{P}) \right\|_{\mathrm{F}} &\le \left\| \left(\boldsymbol{Z}^\top \boldsymbol{Z}\right)^{-1} \right\|_{\mathrm{F}} \cdot \|\boldsymbol{z}_{N+1}\|_2 \cdot \left\| \boldsymbol{t}_{N+1}^\top J_{\boldsymbol{f}}\left(\boldsymbol{P}^\top \boldsymbol{z}_{N+1}\right) \right\|_2 \\
&\quad + \left\| \widetilde{\boldsymbol{Z}}^\top \widetilde{\boldsymbol{G}} \right\|_{\mathrm{F}} \cdot \left\| \frac{\left(\boldsymbol{Z}^\top \boldsymbol{Z}\right)^{-1} \boldsymbol{z}_{N+1} \boldsymbol{z}_{N+1}^\top \left(\boldsymbol{Z}^\top \boldsymbol{Z}\right)^{-1}}{1 + \boldsymbol{z}_{N+1}^\top \left(\boldsymbol{Z}^\top \boldsymbol{Z}\right)^{-1} \boldsymbol{z}_{N+1}} \right\|_{\mathrm{F}}
\end{aligned}$$

In the following, for the sake of simplicity, let $\boldsymbol{A} = \boldsymbol{Z}^\top \boldsymbol{Z}, \boldsymbol{b} = \boldsymbol{z}_{N+1}$ and $\boldsymbol{y} = \boldsymbol{A}^{-1} \boldsymbol{b}$. Then, we have the following inequality with probability at least $1 - \varepsilon$:

$$\left\| \frac{\left(\boldsymbol{Z}^\top \boldsymbol{Z}\right)^{-1} \boldsymbol{z}_{N+1} \boldsymbol{z}_{N+1}^\top \left(\boldsymbol{Z}^\top \boldsymbol{Z}\right)^{-1}}{1 + \boldsymbol{z}_{N+1}^\top \left(\boldsymbol{Z}^\top \boldsymbol{Z}\right)^{-1} \boldsymbol{z}_{N+1}} \right\|_{\mathrm{F}} = \left\| \frac{\boldsymbol{A}^{-1} \boldsymbol{b} \boldsymbol{b}^\top \boldsymbol{A}^{-1}}{1 + \boldsymbol{b}^\top \boldsymbol{A}^{-1} \boldsymbol{b}} \right\|_{\mathrm{F}}$$

$$= \left\| \frac{\left( \boldsymbol{b}^\top \boldsymbol{A}^{-1} \right)^\top \left( \boldsymbol{b}^\top \boldsymbol{A}^{-1} \right)}{1 + \boldsymbol{b}^\top \boldsymbol{A}^{-1} (\boldsymbol{A} \boldsymbol{A}^{-1}) \boldsymbol{b}} \right\|_{\mathrm{F}}$$

$$= \left\| \frac{\boldsymbol{y} \boldsymbol{y}^\top}{1 + \boldsymbol{y}^\top \boldsymbol{A} \boldsymbol{y}} \right\|_{\mathrm{F}}$$

$$= \frac{\|\boldsymbol{y}\|_2^2}{|1 + \boldsymbol{y}^\top \boldsymbol{A} \boldsymbol{y}|}$$

$$\leq \frac{\|\boldsymbol{y}\|_2^2}{\boldsymbol{y}^\top \boldsymbol{A} \boldsymbol{y}}$$

$$\leq \frac{1}{\lambda_{\min}(\boldsymbol{A})} < \zeta,$$

where $\zeta$ is a constant as in Lemma B.1. The first inequality holds since $\boldsymbol{A} \coloneqq \boldsymbol{Z}^\top \boldsymbol{Z}$ is a positive semidefinite matrix with probability $1 - \varepsilon$ by Lemma B.1, and the second inequality follows from the property of Rayleigh quotient. The last inequality directly follows from Lemma B.1. Hence, we have the following inequalities with probability at least $1 - \varepsilon$:

$$\left\| \varphi_{\widetilde{\boldsymbol{T}}}(\boldsymbol{P}) - \varphi_{\boldsymbol{T}}(\boldsymbol{P}) \right\|_{\mathrm{F}} \leq \mu U(\eta + \zeta N),$$

and

$$\left\| \varphi_{\widetilde{\boldsymbol{T}}}(\boldsymbol{P}) - \varphi_{\boldsymbol{T}'}(\boldsymbol{P}) \right\|_{\mathrm{F}} \leq \mu U(\eta + \zeta N).$$

Therefore, we have the following with probability at least $1 - \varepsilon$:

$$\begin{aligned}
\left\| \varphi_{\boldsymbol{T}}(\boldsymbol{P}) - \varphi_{\boldsymbol{T}'}(\boldsymbol{P}) \right\|_{\mathrm{F}} &= \left\| \varphi_{\boldsymbol{T}}(\boldsymbol{P}) - \varphi_{\widetilde{\boldsymbol{T}}}(\boldsymbol{P}) + \varphi_{\widetilde{\boldsymbol{T}}}(\boldsymbol{P}) - \varphi_{\boldsymbol{T}'}(\boldsymbol{P}) \right\|_{\mathrm{F}} \\
&\leq \left\| \varphi_{\widetilde{\boldsymbol{T}}}(\boldsymbol{P}) - \varphi_{\boldsymbol{T}}(\boldsymbol{P}) \right\|_{\mathrm{F}} + \left\| \varphi_{\widetilde{\boldsymbol{T}}}(\boldsymbol{P}) - \varphi_{\boldsymbol{T}'}(\boldsymbol{P}) \right\|_{\mathrm{F}} \\
&\leq 2\mu U(\eta + \zeta N).
\end{aligned}$$

$\square$

## C Proof of Lemma 6.4

*Proof.* Let $\delta^{(i)} \coloneqq \left\| \boldsymbol{P}^{(i)} - \boldsymbol{P}'^{(i)} \right\|_{\mathrm{F}}$. We have $\delta^{(i)} = 0$ for $i = 1, \dots, k$. From Lemma 6.2, we have the following with probability at least $1 - \varepsilon$:

$$\begin{aligned}
\delta^{(i+1)} &= \left\| \boldsymbol{P}^{(i+1)} - \boldsymbol{P}'^{(i+1)} \right\|_{\mathrm{F}} \\
&= \left\| \boldsymbol{P}^{(i+1)} - \boldsymbol{P}^{(i)} + \boldsymbol{P}^{(i)} - \boldsymbol{P}'^{(i)} + \boldsymbol{P}'^{(i)} - \boldsymbol{P}'^{(i+1)} \right\|_{\mathrm{F}} \\
&\leq \left\| \boldsymbol{P}^{(i+1)} - \boldsymbol{P}^{(i)} \right\|_{\mathrm{F}} + \left\| \boldsymbol{P}'^{(i+1)} - \boldsymbol{P}'^{(i)} \right\|_{\mathrm{F}} + \left\| \boldsymbol{P}^{(i)} - \boldsymbol{P}'^{(i)} \right\|_{\mathrm{F}} \\
&\leq 2\eta N U \mu + \delta^{(i)},
\end{aligned}$$

for each $i = k, \dots, K$. Therefore, by using the above relation repeatedly and from Lemma 6.3, we have the following with probability at least $1 - \varepsilon$:

$$\begin{aligned}
\delta^{(K+1)} &\leq 2(K - k)\eta N U \mu + 2\mu U(\eta + \zeta N) \\
&= 2\mu \eta U \left( 1 + \left( K - k + \frac{\zeta}{\eta} \right) N \right),
\end{aligned}$$

which completes the proof. $\square$

## D  Proof of Theorem 6.5

*Proof.* For any $\boldsymbol{t} \in \Delta^{M-1}$ and for any $\{\boldsymbol{T}_i\}_{i=1}^K$, $\{\boldsymbol{T}_i'\}_{i=1}^K \subseteq (\Delta^{M-1})^K$ such that $\{\boldsymbol{T}_i\}_{i=1}^K$ and $\{\boldsymbol{T}_i'\}_{i=1}^K$ differs only one example, we have

$$
\begin{aligned}
\left| \ell(A(\boldsymbol{T}) \,|\, \boldsymbol{t}) - \ell(A(\boldsymbol{T}') \,|\, \boldsymbol{t}) \right| &= \left| \left\| \boldsymbol{b}(\boldsymbol{t} \,|\, \boldsymbol{P}^{(K+1)}) - \boldsymbol{x}^\star(\boldsymbol{t}) \right\|_2^2 - \left\| \boldsymbol{b}(\boldsymbol{t} \,|\, \boldsymbol{P}'^{(K+1)}) - \boldsymbol{x}^\star(\boldsymbol{t}) \right\|_2^2 \right| \\
&\leq \left\| \boldsymbol{b}(\boldsymbol{t} \,|\, \boldsymbol{P}^{(K+1)}) - \boldsymbol{b}(\boldsymbol{t} \,|\, \boldsymbol{P}'^{(K+1)}) \right\|_2^2 \\
&= \left\| \left( \boldsymbol{P}^{(K+1)} - \boldsymbol{P}'^{(K+1)} \right)^\top \boldsymbol{z}(\boldsymbol{t}) \right\|_2^2 \\
&\leq \|\boldsymbol{z}(\boldsymbol{t})\|_2^2 \cdot \left\| \boldsymbol{P}^{(K+1)} - \boldsymbol{P}'^{(K+1)} \right\|_{\mathrm{F}}^2,
\end{aligned}
\tag{16}
$$

where the first inequality follows from the reverse triangle inequality. We can bound the right-hand side of Equation (16) with probability at least $1 - \varepsilon$ by Lemma 6.4. Since the left-hand side of Equation (16) is bounded for all $\boldsymbol{t} \in \Delta^{M-1}$, we see that Algorithm 2 satisfies PAC uniform stability. $\qquad\square$

## E  Problem Definition

**Scaled-MED** is a three-variable three-objective problem defined by:

$$
\begin{aligned}
\text{minimize} \quad & \boldsymbol{f}(\boldsymbol{x}) \coloneqq (f_1(\boldsymbol{x}), f_2(\boldsymbol{x}), f_3(\boldsymbol{x}))^\top \\
\text{subject to} \quad & \boldsymbol{x} \in \mathbb{R}^3 \\
\text{where} \quad & f_1(\boldsymbol{x}) = x_1^2 + 3(x_2 - 1)^2 + 2(x_3 - 1)^2, \\
& f_2(\boldsymbol{x}) = 2(x_1 - 1)^2 + x_2^2 + 3(x_3 - 1)^2, \\
& f_3(\boldsymbol{x}) = 3(x_1 - 1)^2 + 2(x_2 - 1)^2 + (x_3 + 1)^2.
\end{aligned}
$$

**Skew-$M$MED** is an $M$-variable $M$-objective problem defined by:

$$
\begin{aligned}
\text{minimize} \quad & \boldsymbol{f}(\boldsymbol{x}) \coloneqq (f_1(\boldsymbol{x}), \ldots, f_M(\boldsymbol{x}))^\top \\
\text{subject to} \quad & \boldsymbol{x} \in \mathbb{R}^M \\
\text{where} \quad & f_m(\boldsymbol{x}) = \left( \frac{1}{\sqrt{2}} \|\boldsymbol{x} - \boldsymbol{e}_m\|^2 \right)^{p_m}, \\
& p_m = \exp\left( \frac{2(m-1)}{M-1} - 1 \right), \\
& \boldsymbol{e}_m = (0, \ldots, 0, \underbrace{1}_{m\text{-th}}, 0, \ldots, 0)^\top, \\
\text{for} \quad & m = 1, \ldots, M.
\end{aligned}
$$

**Skew-$M$MMD** is an $M$-variable $M$-objective problem defined by:

$$
\begin{aligned}
\text{minimize} \quad & \boldsymbol{f}(\boldsymbol{x}) \coloneqq (f_1(\boldsymbol{x}), \ldots, f_M(\boldsymbol{x}))^\top \\
\text{subject to} \quad & \boldsymbol{x} \in X \subseteq \mathbb{R}^M \\
\text{where} \quad & f_m(\boldsymbol{x}) = \|\boldsymbol{A}_m(\boldsymbol{x} - \boldsymbol{c}_m)\|_2^{p_m}, \\
& p_m > 0, \\
\text{for} \quad & m = 1, \ldots, M.
\end{aligned}
$$

In the experiments in Section 7.2, we set $M = 3$, $X = \mathbb{R}^3$,

$$
\boldsymbol{A}_1 \coloneqq \operatorname{diag}\left( \frac{3}{5}, \frac{4}{5}, \frac{4}{5} \right), \quad \boldsymbol{A}_2 \coloneqq \operatorname{diag}\left( \frac{4}{5}, \frac{3}{5}, \frac{4}{5} \right), \quad \boldsymbol{A}_3 \coloneqq \operatorname{diag}\left( \frac{4}{5}, \frac{4}{5}, \frac{3}{5} \right),
$$

$\boldsymbol{c}_m := \boldsymbol{e}_m$ and $p_m := \exp\left(\frac{2(m-1)}{M-1} - 1\right)$. Note that $\mathrm{diag}\,(\cdot)$ denotes the diagonal matrix.

Regarding the group Lasso instance, we followed the problem definition as in (Tanaka et al., 2020, Appendix E.2.2).

## F  Analytical Solution of Scaled-MED

We derive a map $\boldsymbol{x}^\star \colon \Delta^2 \to X^\star(\boldsymbol{f})$ for scaled-MED. For any $\boldsymbol{t} = (t_1, t_2, t_3)^\top \in \Delta^2$, the scalarizing function weighted by $\boldsymbol{t}$ is defined by

$$
\begin{aligned}
f(\boldsymbol{x} \,|\, \boldsymbol{t}) &:= \sum_{m=1}^{3} t_m f_m(\boldsymbol{x}) \\
&= t_1 x_1^2 + 2t_2(x_1 - 1)^2 + 3t_3(x_1 - 1)^2 \\
&\quad + 3t_1(x_2 - 1)^2 + t_2 x_2^2 + 2t_3(x_2 - 1)^2 \\
&\quad + 2t_1(x_3 - 1)^2 + 3t_2(x_3 - 1)^2 + t_3(x_3 + 1)^2.
\end{aligned}
$$

Since $f(\boldsymbol{x} \,|\, \boldsymbol{t})$ is a convex quadratic function with respect to each $x_1$, $x_2$ and $x_3$, its optimal solution $(x_1^\star(\boldsymbol{t}), x_2^\star(\boldsymbol{t}), x_3^\star(\boldsymbol{t}))^\top$ satisfies the following conditions:

$$
\begin{aligned}
\left.\frac{\partial f(\boldsymbol{x} \,|\, \boldsymbol{t})}{\partial x_1}\right|_{\boldsymbol{x} = \boldsymbol{x}^\star(\boldsymbol{t})} &= 2t_1 x_1 + 4t_2(x_1 - 1) + 6t_3(x_1 - 1) = 0, \\
\left.\frac{\partial f(\boldsymbol{x} \,|\, \boldsymbol{t})}{\partial x_2}\right|_{\boldsymbol{x} = \boldsymbol{x}^\star(\boldsymbol{t})} &= 6t_1(x_2 - 1) + 2t_2 x_2 + 4t_3(x_2 - 1) = 0, \\
\left.\frac{\partial f(\boldsymbol{x} \,|\, \boldsymbol{t})}{\partial x_3}\right|_{\boldsymbol{x} = \boldsymbol{x}^\star(\boldsymbol{t})} &= 4t_1(x_3 - 1) + 6t_2(x_3 - 1) + 2t_3(x_3 + 1) = 0.
\end{aligned}
$$

By solving the above equation, the map $\boldsymbol{x}^\star(\boldsymbol{t})$ is given by

$$
\boldsymbol{x}^\star(\boldsymbol{t}) = (x_1^\star(\boldsymbol{t}), x_2^\star(\boldsymbol{t}), x_3^\star(\boldsymbol{t}))^\top = \left(\frac{2t_2 + 3t_3}{t_1 + 2t_2 + 3t_3}, \frac{3t_1 + 2t_3}{3t_1 + t_2 + 2t_3}, \frac{2t_1 + 3t_2 - t_3}{2t_1 + 3t_2 + t_3}\right)^\top.
$$

## G  Numerical Experiments on Non-simplicial Instances

We present the results of Algorithm 2 for DTLZ instances in Tables 6 and 7. As in Section 7.2, we show the average and the standard deviation of the GDs and IGDs. In this experiment, we picked DTLZ2 and DTLZ4 as non-simplicial instances. The definition of each instance can be found in (Deb et al., 2002, Chapter 6.7). As can be seen from Tables 6 and 7, the proposed method does not work effectively for non-simplicial problems. Thus, the development of optimization methods for non-simplicial problems remains an important direction for future research.

## H  Experiments on Stability

We conducted additional numerical experiments to assess the algorithmic stability. Since directly evaluating stability property is difficult, we examined the empirical difference $\left\|\boldsymbol{P}^{(K+1)} - \boldsymbol{P}'^{(K+1)}\right\|_{\mathrm{F}}$ in Lemma 6.4. To this aim, we repeated the following procedure ten times on the scaled-MED instance:

- Generate two sets of parameters $\boldsymbol{T}_K = \left\{\{\boldsymbol{t}_i^{(1)}\}_{i=1}^N, \ldots, \{\boldsymbol{t}_i^{(k)}\}_{i=1}^N, \ldots, \{\boldsymbol{t}_i^{(K)}\}_{i=1}^N\right\}$ and $\boldsymbol{T}'_K = \left\{\{\boldsymbol{t}_i^{(1)}\}_{i=1}^N, \ldots, \{\boldsymbol{t}_i'^{(k)}\}_{i=1}^N, \ldots, \{\boldsymbol{t}_i^{(K)}\}_{i=1}^N\right\}$ where each $\boldsymbol{t}_i^{(k)}$ ($i = 1, 2, \ldots, N$, $k = 1, 2, \ldots, K$) is drawn uniformly random from the standard simplex $\Delta^{M-1}$. Here, $\{\boldsymbol{t}_i^{(k)}\}_{i=1}^N$ and $\{\boldsymbol{t}_i'^{(k)}\}_{i=1}^N$ are different only one element. In other words, there exist $j \in [N]$ such that $\boldsymbol{t}_j^{(k)} \neq \boldsymbol{t}_j'^{(k)}$ and $\boldsymbol{t}_i^{(k)} \neq \boldsymbol{t}_i'^{(k)}$ for any $i \neq j$.

Table 6: GD and IGD (avg.±s.d. over 100 trials) for DTLZ2.

|  | $N$ | Proposed (Algorithm 2) | NSGAII + all-at-once | MOEA/D + all-at-once |
|---|---|---|---|---|
| GD | 30 | 2.42e+01 ± 1.05e+00 | 5.89e-02 ± 1.93e-03 | **3.66e-02** ± 4.96e-04 |
|  | 50 | 1.82e+01 ± 3.69e-01 | 2.86e-01 ± 5.28e-03 | **3.44e-02** ± 4.35e-04 |
|  | 100 | 1.47e+01 ± 2.08e-01 | 4.56e-02 ± 3.26e-04 | **2.98e-02** ± 3.56e-04 |
| IGD | 30 | 1.67e+00 ± 6.27e-02 | **2.51e-01** ± 5.59e-04 | 3.83e-01 ± 1.83e-04 |
|  | 50 | 3.20e+00 ± 2.28e-01 | 4.93e-01 ± 6.77e-05 | **4.42e-01** ± 1.12e-03 |
|  | 100 | 1.54e+00 ± 2.64e-01 | 5.89e-01 ± 1.40e-03 | **2.85e-01** ± 5.71e-04 |

Table 7: GD and IGD (avg.±s.d. over 100 trials) for DTLZ4.

|  | $N$ | Proposed (Algorithm 2) | NSGAII + all-at-once | MOEA/D + all-at-once |
|---|---|---|---|---|
| GD | 30 | **4.34e-02** ± 1.56e-03 | 1.93e+01 ± 4.39e-01 | 6.83e-02 ± 1.75e-03 |
|  | 50 | **4.67e-02** ± 1.70e-03 | 3.54e+00 ± 9.42e-02 | 3.68e-01 ± 5.97e-03 |
|  | 100 | **5.05e-02** ± 1.86e-03 | 4.55e+02 ± 1.88e+01 | 4.07e-01 ± 5.90e-03 |
| IGD | 30 | 1.97e+00 ± 2.13e-04 | 8.52e-02 ± 2.80e-02 | **4.07e-02** ± 2.69e-04 |
|  | 50 | 1.97e+00 ± 5.99e-04 | 6.14e-02 ± 9.07e-03 | **4.76e-02** ± 5.80e-03 |
|  | 100 | 1.97e+00 ± 5.72e-04 | 6.51e-02 ± 2.75e-02 | **5.41e-02** ± 8.32e-03 |

Table 8: Empirical difference $\left\|\boldsymbol{P}^{(K+1)} - \boldsymbol{P}'^{(K+1)}\right\|_{\mathrm{F}}$ for $k \in \{20, 40, 60, 80\}$.

|  | $k = 20$ | $k = 40$ | $k = 60$ | $k = 80$ |
|---|---|---|---|---|
| $\|\boldsymbol{P}^{(K+1)} - \boldsymbol{P}'^{(K+1)}\|_{\mathrm{F}}$ | 1.21e-03 | 1.78e-03 | 1.10e-03 | 1.09e-03 |
|  | ± 5.13e-04 | ± 1.50e-03 | ± 4.89e-04 | ± 3.16e-04 |

- Run the proposed method (Algorithm 2) with a maximum number of iterations $K = 100$ for $T_K$ and $T'_K$ and obtain the resulting control points matrix $\boldsymbol{P}^{(K+1)}$ and $\boldsymbol{P}'^{(K+1)}$, respectively.

- Calculate the difference $\|\boldsymbol{P}^{(K+1)} - \boldsymbol{P}'^{(K+1)}\|_{\mathrm{F}}$.

We show the results of the average and standard deviation of $\|\boldsymbol{P}^{(K+1)} - \boldsymbol{P}'^{(K+1)}\|_{\mathrm{F}}$ for $k \in \{20, 40, 60, 80\}$ below. From this table, we see that the average of the empirical difference did not necessarily decrease for large $k$, while its upper bound is $O(1/k)$ in Lemma 6.4. However, the scale of difference was sufficiently small in the absolute sense, even for the smallest $k = 20$. Thus, while there may be a gap between the empirical difference $\left\|\boldsymbol{P}^{(K+1)} - \boldsymbol{P}'^{(K+1)}\right\|_{\mathrm{F}}$ and the theoretical upper bound, the proposed method are expected to be stable in practice.

