# OpenReview forum: "Bézier Flow: a Surface-wise Gradient Descent Method for Multi-objective Optimization"
_TMLR — Accepted by TMLR_

### Review · Reviewer_Df3Q · 2024-11-05

**Summary Of Contributions:**

- This work proposes a multi-objective optimization algorithm: it is the first method that directly operates on a parametrization of the Pareto set instead of working with a finite set of samples.
- It then extends the stability of optimization algorithms to this setting and establishes generalization bounds for their algorithm.

**Audience:**

Yes

**Claims And Evidence:**

Yes

**Requested Changes:**

- Discuss the "decomposability" issue.
- Discuss the computational overhead.

**Strengths And Weaknesses:**

# Strengths
- Novel algorithm that provides significant conceptual and practical improvments
- Novel generalization bounds for multi objective optimization algoorithms

# Weaknesses
## PAC stability and generalization
Though the motivation for the PAC stability framework is to obtain generalization bounds, the effective obtention of such bounds for Alg. 2 is conditional on Alg. 2 being "decomposable", which does not seem trivial to me (cf remark after Thm. 6.5).

## Numerical experiments
There is no mention of the computational overhead of the proposed method. Instead of performing a Bezier fit once like existing methods, this algorithm performs one at every iteration. It would be fairer to compare methods for a given computational budget, or at least report CPU times.

---

> ### Author Response · Authors · 2024-12-29
>
> Thank you for your careful review and insightful comments. We are grateful for your positive and constructive feedback. Our point-by-point responses to the reviewer's comments are described below.
>
> At this time, we have not yet received comments from all reviewers, and have not revised the manuscript yet. However, we will incorporate all reviewers' feedback and additional results into the revised manuscript.
>
>
> > #### PAC stability and generalization
> > Though the motivation for the PAC stability framework is to obtain generalization bounds, the effective obtention of such bounds for Alg. 2 is conditional on Alg. 2 being "decomposable", which does not seem trivial to me (cf remark after Thm. 6.5).
>
> Thank you for your comment. As the reviewer pointed out, it is indeed not trivial to determine whether an algorithm is decomposable is not in practice. However, the decomposability is just a theoretical assumption, and we believe that this is not restrictive in practice for the following reason.
>
> Suppose that an algorithm $\mathcal A$ is not decomposable, meaning that there exists $\overline{\varepsilon}\in (0,1)$ such that there are no event $B_{\bar{\varepsilon}}$ satisfying $D_{\bar{\varepsilon}}=B_{\bar{\varepsilon}}^{N+1}$.  Even in this case, we can often take a smaller $\varepsilon' < \overline{\varepsilon}$ so that there exists an event $B_{\varepsilon'}$ satisfying $B_{\varepsilon'}^{N+1}\subseteq D_{\bar{\varepsilon}}$.
> Thus, by letting an event as $D_{\varepsilon'}\leftarrow B_{\varepsilon'}^{N+1}$, we can bound the difference between the generalization and empirical errors with an event $D_{\varepsilon'}$ which occurs with a certain probability.
>
> > #### Numerical experiments
> > There is no mention of the computational overhead of the proposed method. Instead of performing a Bezier fit once like existing methods, this algorithm performs one at every iteration. It would be fairer to compare methods for a given computational budget, or at least report CPU times.
>
> Thank you for your suggestion. To address the computational budget, we measured entire computation time of our method and the baselines (NSGA-II and MOEA/D with All-at-Once) for skewMED and skewMMD. The tables show the average and standard deviation of computation time in seconds of each method over 10 trials.
>
> - Computation time (sec) for skewMED
>
> | | Proposed | NSGA-II + All-at-Once | MOEA/D + All-at-Once |
> | --- | --- | --- | --- |
> | $N=30$ | 1.09e+01 $\pm$ 5.79e-01 | 1.41e-00 $\pm$ 1.09e-01 | 1.92e+00 $\pm$ 4.92e-02 |
> | $N=50$ | 1.07e+01 $\pm$ 7.25e-01 | 1.67e-00 $\pm$ 4.06e-02 | 2.04e+00 $\pm$ 5.38e-02 |
> | $N=100$ | 1.08e+01 $\pm$ 6.72e-01 | 2.57e-00 $\pm$ 8.76e-02 | 1.85e+00 $\pm$ 3.74e-02 |
>
> - Computation time (sec) for skewMMD
>
> |  | Proposed | NSGA-II + All-at-Once | MOEA/D + All-at-Once |
> | --- | --- | --- | --- |
> | $N=30$ | 1.32e+01 $\pm$ 7.26e-01 | 1.39e-00 $\pm$ 8.89e-03 | 2.04e+00 $\pm$ 5.38e-02 |
> | $N=50$ | 1.29e+01 $\pm$ 5.84e-01 | 1.83e-00 $\pm$ 9.01e-02 | 2.12e+00 $\pm$ 2.34e-02 |
> | $N=100$ | 1.30e+01 $\pm$ 6.95e-01 | 2.69e-00 $\pm$ 7.59e-02 | 2.43e+00 $\pm$ 2.89e-02 |
>
>
> From these tables, we see that our methods took about 10 seconds to output the resulting Bezier simplex, whereas the post-processing method (NSGA-II and MOEA/D with Bezier simplex fitting) needed only about 1–2 seconds for each $N$. This is mainly because our method performs Bezier simplex fitting at every iteration, whereas existing methods do this only once.
>
> On the other hand, the computation time for our methods remained almost the same even when $N$ increases 100, while it grows for the existing methods. Although our method was slower in our experiments, their computing time did not increase much with larger $N$, which can be advantageous in certain situations.

---

### Review · Reviewer_tznU · 2024-11-07

**Summary Of Contributions:**

Overview:

This paper constructs an algorithm to convert a single objective optimization to a multiple
objective optimization and examine the PAC stability (and thus upper bound for the generalization gap).

**Audience:**

Yes

**Claims And Evidence:**

Yes

**Requested Changes:**

Please address the weaknesses listed above.
* verification of the stability empirically
* experimental results on real world data
* show clear advantages over baseline methods

**Strengths And Weaknesses:**

Strengths:
+ A new algorithm is built to transform a single-objective optimization
into a multi-objective optimization problem.

+ Theoretical properties such as PAC stability of the algorithm is studied.

Weaknesses:
- Only a simple numerical simulation is performed to compare the proposed method.
No real world data is used for comparison.

- It would be desirable if the stability property can be verified with the simulation or
empirical results.

- The results in Table 3 appear to show that the proposed method and the baseline have comparable accuracy
and convergence speed. The advantage of the proposed method does not appear to be
so much.

---

> ### Author Response · Authors · 2024-12-29
> **Numerical experiments on practical situation**
>
> Thank you for your careful review and insightful comments. We are grateful for your positive and constructive feedback. Our point-by-point responses to the reviewer's comments are described below.
>
> At this time, we have not yet received comments from all reviewers, and have not revised the manuscript yet. However, we will incorporate all reviewers' feedback and additional experimental results into the revised manuscript.
>
> > Only a simple numerical simulation is performed to compare the proposed method. No real world data is used for comparison.
>
> We appreciate your comments. To compare the performance of our method and existing ones under more practical situations, we additionally applied these methods to the multi-objective hyper-parameter tuning problem of the group Lasso, formulated by (Tanaka et al., 2020). Following (Tanaka et al., 2020), we used the Birthwt dataset, which consists of 189 births at the Baystate Medical Centre, Springfield, Massachusetts, in 1986 (Hosmer et al., 2013). From this dataset, we used six continuous features `age1`, `age2`, `age3`, `lwt1`, `lwt2`, `lwt3` (mother's age in years and mother's weight in pounds at the last menstrual period) as predictors and one continuous feature `bwt` (birth weight in grams) as a response for regression.
>
> Let $A\in \mathbb R^{189\times 6}$ be a matrix of observations of the predictors. Let $\boldsymbol{x} = (x_1,x_2,\dots,x_6)^{\top} \in \mathbb{R}^6$ be a vector of the predictor coefficients to be estimated, separated into to groups, $\boldsymbol{x}\_{\text{age}}=(x\_1, x\_2, x\_3)\^\top$ and $\boldsymbol{x}_{\text{lwt}}=(x_4, x_5, x_6)^\top$, and $\boldsymbol{y}\in\mathbb{R}^{189}$ be a vector of observations the response.
>
> Tanaka et al. (2020) formulated the group Lasso problem for linear regression as a multi-objective optimization problem with three objectives: minimizing the mean squared error, the $\ell_2$ norm of the `age` group, and the $\ell_2$ norm of the `lwt` group. The problem is to solve the following three-objective optimization problem:
>
> $$
> \min\_{x\in\mathbb{R}\^6}~(\tilde{f}\_1(x), \tilde{f}\_2(x), \tilde{f}\_3(x))\^{\top},
> $$
>
> where each objective function is defined as follows:
>
> \begin{align}
> \tilde{f}\_1(\boldsymbol{x}) &= \frac{1}{189}\\|A\boldsymbol{x} - \boldsymbol{y}\\|\^2\_2 + \varepsilon\\|\boldsymbol{x}\\|\^2\_2, \\\\
> \tilde{f}\_2(\boldsymbol{x}) &= \\|\boldsymbol{x}\_{\text{age}}\\|\_2 + \varepsilon\\|\boldsymbol{x}\\|\^2\_2, \\\\
> \tilde{f}\_3(\boldsymbol{x}) &= \\|\boldsymbol{x}\_{\text{lwt}}\\|\_2 + \varepsilon\\|\boldsymbol{x}\\|\^2\_2,
> \end{align}
>
> where $\\|\cdot\\|\_2$ is the $\ell_2$ norm on an appropriate space, and $\varepsilon>0$ is an arbitrarily small positive number which ensures that each objective function $\tilde{f}_1, \tilde{f}_2,$ and $\tilde{f}_3$ is strongly convex. In the experiments, we set $\varepsilon := 10^{-4}$.
>
> We applied our method (Algorithm 2) and the existing multi-objective algorithms (NSGA-II and MOEA/D) with Bezier simplex fitting (All-at-Once) to this problem and compared their performance in terms of GD (Generational Distance) and IGD (Inverted Generational Distance). To calculate GD and IGD, we used the approximate Pareto solutions obtained by NSGA-III (Deb and Jain, 2014) with a population size of 1000 and a maximum of 200,000 evaluations as a validation set $Y$. The other settings were the same as in Section 7.2 of the original manuscript.
>
> We report the average and standard deviation of GD and IGD of our method (Algorithm2) and the existing post-processing methods (NSGA-II and MOEA/D with All-at-Once method) over 100 trials in the following tables. These tables show that our method achieved better GD and IGD than post-processing methods. These results demonstrate the potential that our method provides better performance than existing methods, even in real-world scenarios. We hope this additional evidence clarifies the practical merits of our approach.
>
> - GD for the group Lasso
> 	|  | proposed | NSGA-II + All-at-Once | MOEA/D + All-at-Once |
> 	| --- | --- | --- | --- |
> 	| $N=30$ | **2.44e-02** $\pm$ 5.25e-04 | 6.47e-02 $\pm$ 1.75e-04 | 6.78e-02 $\pm$ 2.78e-03 |
> 	| $N=50$ | **2.37e-02** $\pm$ 4.72e-04 | 4.29e-02 $\pm$ 3.73e-04 | 4.13e-02 $\pm$ 4.51e-04 |
> 	| $N=100$ | **2.37e-02** $\pm$ 4.30e-04 | 4.56e-02 $\pm$ 5.06e-04 | 4.12e-02 $\pm$ 3.10e-04 |
> - IGD of the group Lasso
> 	| population size | proposed | NSGA-II | MOEA/D |
> 	| --- | --- | --- | --- |
> 	| $N=30$ | **5.94e-02** $\pm$ 3.98e-04 | 2.05e-01 $\pm$ 4.12e-04 | 1.35e-01 $\pm$ 2.76e-04 |
> 	| $N=50$ | **5.95e-02** $\pm$ 3.79e-04 | 1.18e-01 $\pm$ 3.99e-04 | 9.96e-02 $\pm$ 3.97e-04 |
> 	| $N=100$ | **5.92e-02** $\pm$ 3.82e-04 | 1.24e-01 $\pm$ 5.13e-04 | 6.44e-02 $\pm$ 3.12e-04 |
>
> By the way, please note that we used NSGA-III to generate a validation set in this additional experiment while we employed NSGA-II in our original manuscript. The reason why we used NSGA-III is described in our response to the last comment.

---

> > ### Author Response · Authors · 2024-12-29
> > **Stability experiment**
> >
> > > It would be desirable if the stability property can be verified with the simulation or empirical results.
> >
> > Following the reviewer's suggestion, we conducted additional numerical experiments to assess the algorithmic stability. Since directly evaluating stability is difficult, we examined the empirical difference $\left\\|\boldsymbol{P}^{(K+1)}-\boldsymbol{P}\^{\prime(K+1)}\right\\|\_{\mathrm{F}}$ in Lemma 6.4.
> > To this aim, we repeated the following procedure 10 times to the scaled-MED instance:
> >
> > - Generate two sets of parameters $T\_K = \\{\\{\boldsymbol{t}\^{(1)}\_i\\}\_{i=1}\^{N},\dots,\\{\boldsymbol{t}\^{(k)}\_i\\}\_{i=1}\^{N},\dots,\\{\boldsymbol{t}\^{(K)}\_i\\}\_{i=1}\^{N}\\}$ and $T'\_K = \\{\\{\boldsymbol{t}\^{(1)}\_i\\}\_{i=1}\^{N},\dots,\\{\boldsymbol{t}'\^{(k)}\_i\\}\_{i=1}\^{N},\dots,\\{\boldsymbol{t}\^{(K)}\_i\\}\_{i=1}\^{N}\\}$ where each $\boldsymbol{t}_i^{(k)}$ ($i=1,2,\dots,N$, $k=1,2,\dots,K$) is drawn uniformly random from the standard simplex $\Delta\^{M-1}$. Here, $\\{\boldsymbol{t}\^{(k)}\_i\\}\_{i=1}\^{N}$ and $\\{\boldsymbol{t}'\^{(k)}\_i\\}\_{i=1}\^{N}$ are different only one element. In other words, there exist $j\in[N]$ such that $t^{(k)}_j \neq t'^{(k)}_j$ and $t^{(k)}_i \neq t'^{(k)}_i$ for any $i\neq j$.
> > - Run the proposed method (Algorithm 2) with a maximum number of iterations $K=100$ for $T_K$ and $T'_K$ and obtain the resulting control points matrix $\boldsymbol{P}^{(K+1)}$ and $\boldsymbol{P}'^{(K+1)}$, respectively.
> > - Calculate the difference $\\|\boldsymbol{P}^{(K+1)} - \boldsymbol{P}'^{(K+1)}\\|_{\text{F}}$.
> >
> > We show the results of the average and standard deviation of $\\|\boldsymbol{P}\^{(K+1)} - \boldsymbol{P}'\^{(K+1)}\\|\_{\text{F}}$ for $k\in\\{20,40,60,80\\}$ below. From this table, we see that the average of the empirical difference did not necessarily decrease for large $k$, while its upper bound is $O(1/k)$ in Lemma 6.4. However, the scale of difference was sufficiently small in the absolute sense, even for the smallest $k=20$. Thus, while there may be a gap between the empirical difference $\left\\|\boldsymbol{P}^{(K+1)}-\boldsymbol{P}^{\prime(K+1)}\right\\|\_{\mathrm{F}}$ and the theoretical upper bound, the proposed method are expected to be stable in practice.
> >
> > |  | $k=20$ | $k=40$ | $k=60$ | $k=80$ |
> > | --- | --- | --- | --- | --- |
> > |  $\lVert\boldsymbol{P}\^{(K+1)} - \boldsymbol{P}\^{'(K+1)}\rVert\_{\mathrm{F}}$ | 1.21e-03 $\pm$ 5.13e-04 | 1.78e-03 $\pm$ 1.50e-03 | 1.10e-03 $\pm$ 4.89e-04 | 1.09e-03 $\pm$ 3.16e-04 |

---

> > > ### Author Response · Authors · 2024-12-29
> > > **Comparison with baseline and Reference**
> > >
> > > > The results in Table 3 appear to show that the proposed method and the baseline have comparable accuracy and convergence speed. The advantage of the proposed method does not appear to be so much.
> > >
> > > Thank you for your comments. In the original manuscript, we used the approximate Pareto solutions obtained from NSGA-II as the validation set for the computation of GD and IGD. However, we realized after the submission that this could potentially be biased in favor of NSGA-II when evaluating its performance. To address this issue and ensure a more objective comparison, we re-evaluated GD and IGD using a validation set constructed from the output of NSGA-III instead of NSGA-II. We ran NSGA-III with a population size of 1000 and a maximum of 200,000 evaluations.
> > >
> > > - GD for skewMMD
> > > 	| | Proposed | NSGA-II + All-at-Once | MOEA/D + All-at-Once |
> > > 	| --- | --- | --- | --- |
> > > 	| $N=30$ | **7.07e-02** $\pm$ 2.05e-03 | 2.59e-01 $\pm$ 7.41e-03 | 1.57e-01 $\pm$ 5.42e-03 |
> > > 	| $N=50$ | **6.53e-02** $\pm$ 1.18e-03 | 1.26e-01 $\pm$ 4.72e-03 | 9.36e-02 $\pm$ 1.46e-03 |
> > > 	| $N=100$ | **6.59e-02** $\pm$ 1.23e-03 | 7.57e-02 $\pm$ 1.11e-03 | 6.63e-02 $\pm$ 9.19e-04 |
> > >
> > > - IGD for skewMED
> > > 	| | Proposed | NSGA-II + All-at-Once | MOEA/D + All-at-Once |
> > > 	| --- | --- | --- | --- |
> > > 	| $N=30$ | **9.91e-02** $\pm$ 8.43e-04 | 1.69e-01 $\pm$ 5.39e-03 | 1.28e-01 $\pm$ 3.35e-03 |
> > > 	| $N=50$ | **9.75e-02** $\pm$ 4.84e-04 | 1.15e-01 $\pm$ 1.89e-02 | 1.09e-01 $\pm$ 3.47e-03 |
> > > 	| $N=100$ | **8.98e-02** $\pm$ 5.22e-04 | 9.14e-02 $\pm$ 3.44e-03 | 9.77e-02 $\pm$ 2.78e-03 |
> > >
> > > - GD for skewMMD
> > > 	| | Proposed | NSGA-II + All-at-Once | MOEA/D + All-at-Once |
> > > 	| --- | --- | --- | --- |
> > > 	| $N=30$ | **5.57e-02** $\pm$ 1.69e-03 | 1.61e-01 $\pm$ 3.19e-03 | 5.94e-02 $\pm$ 9.98e-04 |
> > > 	| $N=50$ | **5.33e-02** $\pm$ 1.25e-03 | 9.98e-02 $\pm$ 1.98e-03 | 5.94e-02 $\pm$ 9.86e-04 |
> > > 	| $N=100$ | **5.14e-02** $\pm$ 1.02e-03 | 8.57e-02 $\pm$ 2.50e-02 | 5.94e-02 $\pm$ 2.54e-03 |
> > >
> > > - IGD for skewMMD
> > > 	| | Proposed | NSGA-II + All-at-Once | MOEA/D + All-at-Once |
> > > 	| --- | --- | --- | --- |
> > > 	| $N=30$ | **6.69e-02** $\pm$ 6.73e-04 | 8.57e-02 $\pm$ 2.78e-03 | 8.58e-02 $\pm$ 3.16e-03 |
> > > 	| $N=50$ | **6.34e-02** $\pm$ 5.57e-04 | 7.17e-02 $\pm$ 2.48e-03 | 6.45e-02 $\pm$ 3.06e-03 |
> > > 	| $N=100$ | **6.28e-02** $\pm$ 4.62e-04 | 1.01e-01 $\pm$ 6.31e-02 | 7.31e-02 $\pm$ 2.95e-03 |
> > >
> > > The results show that contrary to the initial findings from Table 3 in the original manuscript, the proposed methods outperformed the existing approaches in both GD and IGD for skew-MED and skew-MMD.
> > >
> > > In addition, we would like to emphasize that our method offers theoretical advantages, particularly the PAC stability property, which is not guaranteed by traditional evolutionary methods. Our methods not only achieve comparable performance to existing approaches but also provide a stability guarantee. In this sense, we believe that our methods constitute a valuable contribution to multi-objective optimization research.
> > >
> > >
> > > ### Reference
> > >
> > > - (Deb and Jain, 2014) Deb, K., & Jain, H. (2014). An evolutionary many-objective optimization algorithm using reference-point based non-dominated sorting approach, Part I: Solving problems with box constraints. IEEE Transactions on Evolutionary Computation, 18(4), 577–601.
> > > - (Hosmer et al.,2013) Hosmer Jr., D. W., Lemeshow, S., & Sturdivant, R. X. (2013). Applied Logistic Regression. 3rd ed., Wiley Series in Probability and Statistics. John Wiley & Sons.
> > > - (Tanaka et al., 2020) Tanaka, A., Sannai, A., Kobayashi, K., & Hamada, N. (2020). Asymptotic risk of Bézier simplex fitting. In Proceedings of the AAAI Conference on Artificial Intelligence, 2416–2424.

---

### Review · Reviewer_X4ks · 2024-12-29

**Summary Of Contributions:**

This paper presents an approach for fitting Pareto frontier (parametric) surrogate functions in multi-objective optimization. In particular, it builds on Bezier simplexes with a simple algorithm that alternates between sampling points from a currently-fit simplex, optimizing (or otherwise updating) these points, and fitting a simplex to the (updated) points. The authors further discuss a version of this approach in which gradient descent is used to optimize individual points, and present an update algorithm for simplex parameters. The authors present a PAC stability analysis of the methods, which appears correct. The authors also include numerical experiments. I am not an expert in multi-objective optimization and so I do not know if the scope and scale of these experiments is reasonable for the subfield.

**Audience:**

Yes

**Broader Impact Concerns:**

This is a narrowly technical paper that I do not believe requires a broader impact statement.

**Claims And Evidence:**

Yes

**Requested Changes:**

As I mentioned previously, I believe this paper is quite close to ready for publication. Please address the questions in my weaknesses section above and, if possible, include the experiments mentioned and if reasonable.

**Strengths And Weaknesses:**

# Strengths
- To the best of my knowledge, the approach is reasonably novel. The framework is general-purpose and the choice of Bezier simplexes seems well motivated.
- The theoretical guarantees of the proposed method are a strong contribution.
- Overall, the paper presents a reasonable method for a particular class of problems, and is very close to publication ready. The paper is quite clear and reasonably self-contained.

# Weaknesses
- The proposed approach is limited to simplicial problems. This is not a major weaknesses, as it is reasonable to restrict the set of problems of interest and exploit the structure of those problems.
- The experimental evaluation seems (to a non-expert in this subfield) to be quite limited. I will leave it to the other reviewers to comment on whether the current set of experiments is comprehensive enough for publication. However, related to the last point: it would be interesting to experimentally characterize the performance on non-simplicial problems. Are such experiments possible?
- As far as I can tell, there was no analysis of the scalability of the method. How many points are reasonable to define the hypersurface (in varying dimensions)? What is the computational complexity of the simplex fitting procedure? Are there any other factors impacting the scalability of the method over alternate approaches?
- Could the gradient descent method be extended to multiple steps of gradient descent? If so, it could enable re-use of previous {x} in modeling, if this is useful. How would this impact the computation complexity of Bezier simplex parameter updates? Moreover, could this provide a bridge between sampling-based Pareto front computation and your approach?

---

> ### Author Response · Authors · 2025-01-06
>
> Thank you for your careful review and insightful comments. We are grateful for your positive and constructive feedback. Our point-by-point responses to the reviewer's comments are described below.
>
> > The experimental evaluation seems (to a non-expert in this subfield) to be quite limited. I will leave it to the other reviewers to comment on whether the current set of experiments is comprehensive enough for publication. However, related to the last point: it would be interesting to experimentally characterize the performance on non-simplicial problems. Are such experiments possible?
>
> Thank you for your suggestion. However, due to the time constraint, it would be difficult to complete the numerical experiments on non-simplicial problem instances in about two weeks. If our manuscript is accepted, we would like to include the experiments on non-simplicial problems and discuss the results in the final version.
>
> > As far as I can tell, there was no analysis of the scalability of the method. How many points are reasonable to define the hypersurface (in varying dimensions)?
>
> The required number of points must be at least equal to the number of control points of the Bézier simplex to be estimated. Specifically, if the degree of Bézier simplex is $D$ and the number of objective functions is $M$, it must be at least $|\mathbb{N}^M_D| = \binom{D+M-1}{D} = O(M^D)$ number of points. Thus, the number of required points grows exponentially with respect to $M$.
>
> > What is the computational complexity of the simplex fitting procedure?
>
> The computational complexity of the simplex fitting is essentially equivalent to the complexity of updating control points in Eq. (6). Specifically, since  $Z^{(k)}\in \mathbb R^{|\mathbb{N}^M_D|\times |\mathbb{N}^M_D|}$, the complexity for updating control points is $O(|\mathbb{N}^M_D|^3)$.
>
> > Are there any other factors impacting the scalability of the method over alternate approaches?
>
> Another factor of the scalability of the proposed method is the degree $D$ of the Bézier simplex, which does not appear in other approaches that do not use Bézier simplices in multi-objective optimization.
>
> > Could the gradient descent method be extended to multiple steps of gradient descent? If so, it could enable re-use of previous {x} in modeling, if this is useful. How would this impact the computation complexity of Bezier simplex parameter updates? Moreover, could this provide a bridge between sampling-based Pareto front computation and your approach?
>
> Thank you for your question. It is possible to extend our method to multi-step settings, while currently, our method updates the control points to fit a Bézier simplex after a single-step gradient descent.
>
> The overall computation complexity depends on the number of times control points are updated. Thus, if multi-step gradient descent accelerates the convergence speed of the control points, it may contribute to reducing the computation time of our method, which is an interesting direction for further research.
>
> > Moreover, could this provide a bridge between sampling-based Pareto front computation and your approach?
>
> We reply to this comment assuming that the reviewer is asking about the sampling strategy for the parameters. While we are not sure whether the multi-step gradient descent can directly provide an insight into sampling parameters $\\{\boldsymbol{t}_i\\}\_{i=1}\^{N} \subseteq \Delta\^{M-1}$, we believe that existing sampling strategies for Pareto-front computation can be used in our method. For example, Tanaka et al. (2020) showed that, in some cases, stratified sampling from each skeleton of a simplex can reduce the asymptotic risk of Bézier simplex fitting for describing the Pareto front. Thus, by using such strategies, we may update the Bézier simplex with a small number of samples.
>
> If we have misunderstood the reviewer's question, please let us know. We would be happy to discuss further.
>
> ### Reference
>
> (Tanaka et al., 2020) Tanaka, A., Sannai, A., Kobayashi, K., & Hamada, N. (2020). Asymptotic risk of Bézier simplex fitting. In Proceedings of the AAAI Conference on Artificial Intelligence, 2416–2424.

---

> > ### Author Response · Authors · 2025-03-11
> > **Experiments to non-simplicial problems**
> >
> > We appreciate your significant and intriguing suggestion. We have applied the proposed method to DTLZ2 and DTLZ4, which are widely used benchmarks in the multi-objective optimization community, as non-simplicial multi-objective optimization problems. Similar to experiments on other multi-objective optimization problems in the manuscript, we calculated GD/IGD and compared the results with existing fitting methods. Below, we report the results.
> >
> > - GD of DTLZ2
> >
> > |  | Proposed | NSGA-II + All-at-Once | MOEA/D + All-at-Once |
> > | --- | --- | --- | --- |
> > |N=30 |  2.42e+01 ± 1.05e+00 | 5.89e-02 ± 1.93e-03 | 3.66e-02 ± 4.96e-04 |
> > |N=50 |  1.82e+01 ± 3.69e-01 | 2.86e-01 ± 5.28e-03 | 3.44e-02 ± 4.35e-04 |
> > |N=100|  1.47e+01 ± 2.08e-01 | 4.56e-02 ± 3.26e-04 | 2.98e-02 ± 3.56e-04 |
> >
> > - IGD of DTLZ2
> >
> > |  | Proposed | NSGA-II + All-at-Once | MOEA/D + All-at-Once |
> > | --- | --- | --- | --- |
> > |N=30 |  1.67e+00 ± 6.27e-02 | 2.51e-01 ± 5.59e-04 | 3.83e-01 ± 1.83e-04 |
> > |N=50 |  3.20e+00 ± 2.28e-01 | 4.93e-01 ± 6.77e-05 | 4.42e-01 ± 1.12e-03 |
> > |N=100|  1.54e+00 ± 2.64e-01 | 5.89e-01 ± 1.40e-03 | 2.85e-01 ± 5.71e-04 |
> >
> > - GD of DTLZ4
> >
> > |  | Proposed | NSGA-II + All-at-Once | MOEA/D + All-at-Once |
> > | --- | --- | --- | --- |
> > | N=30 |  4.34e-02 ± 1.56e-03 |  1.93e+01 ± 4.39e-01 |  6.83e-02 ± 1.75e-03 |
> > | N=50 |  4.67e-02 ± 1.70e-03 | 3.54e+00 ± 9.42e-02 | 3.68e-01 ± 5.97e-03 |
> > | N=100|  5.05e-02 ± 1.86e-03 | 4.55e+02 ± 1.88e+01 | 4.07e-01 ± 5.90e-03 |
> >
> > - IGD of DTLZ4
> >
> > |  | Proposed | NSGA-II + All-at-Once | MOEA/D + All-at-Once |
> > | --- | --- | --- | --- |
> > | N=30 |  1.97e+00 ± 2.13e-04 |  8.52e-02 ± 2.80e-02 | 4.07e-02 ± 2.69e-04 |
> > | N=50 |  1.97e+00 ± 5.99e-04 | 6.14e-02 ± 9.07e-03 | 4.76e-02 ± 5.80e-03 |
> > | N=100|  1.97e+00 ± 5.72e-04 | 6.51e-02 ± 2.75e-02 | 5.41e-02 ± 8.32e-03 |
> >
> > As can be seen from these results, the proposed method does not perform effectively for non-simplicial problems. This study focuses on simplicial problems, and the development of optimization methods for non-simplicial problems remains an important direction for future research.

---

### Decision · Action_Editor_NQpA · 2025-03-04

**Recommendation:** Accept as is

**Comment:**

This manuscript presents a novel approach to extend a single-objective optimization algorithm to the multi-objective setting by appealing to a tool known as the Bézier simplex model and provide an analysis of this approach in the PAC framework.

The paper received a positive reception during the initial review phase, with the reviewers only raising some minor questions and comments. The discussion phase served to clear up these minor issues, which were satisfactorily addressed by the authors in their comments and revision.

After the discussion / revision phase, the reviewers are unanimous in their support of the paper being accepted at TMLR. In particular, there is broad agreement that the claims made in the paper are supported by clear and convincing evidence and that the content of the paper would be of interest to a segment of TMLR's target audience.

**Audience:**

Yes, this paper presents a new strategy for multi-objective optimization, a topic of interest to a nontrivial segment of TMLR's target audience.

**Claims And Evidence:**

Yes, after the discussion / revision phase, the reviewers agree that the claims made in the submission are supported by accurate, convincing, and clear evidence.